biomathematics

COVID-19, immunology and inflammation, mathematical model, T cell

**Authors for correspondence:**
Gregory D. Scholes
e-mail: gscholes@princeton.edu
Zhen-Su She
e-mail: she@pku.edu.cn

[†]These authors contributed equally.

# Impairment of T cells' antiviral and anti-inflammation immunities may be critical to death from COVID-19

Luhao Zhang[1,3,†], Rong Li[1,2,†], Gang Song[4],
Gregory D. Scholes[3] and Zhen-Su She[1,2]

[1]Institute of Health System Engineering, College of Engineering, and [2]State Key Laboratory for Turbulence and Complex Systems, Peking University, Beijing 100871, People's Republic of China
[3]Department of Chemistry, Princeton University, Princeton, NJ 08540, USA
[4]Beijing Hospital, National Center of Gerontology, Institute of Geriatric Medicine, Chinese Academy of Medical Sciences, Beijing 100730, People's Republic of China

GDS, 0000-0003-3336-7960; Z-SS, 0000-0001-7001-9995

Clarifying dominant factors determining the immune heterogeneity from non-survivors to survivors is crucial for developing therapeutics and vaccines against COVID-19. The main difficulty is quantitatively analysing the multi-level clinical data, including viral dynamics, immune response and tissue damages. Here, we adopt a top-down modelling approach to quantify key functional aspects and their dynamical interplay in the battle between the virus and the immune system, yielding an accurate description of real-time clinical data involving hundreds of patients for the first time. The quantification of antiviral responses gives that, compared to antibodies, T cells play a more dominant role in virus clearance, especially for mild patients (96.5%). Moreover, the anti-inflammatory responses, namely the cytokine inhibition and tissue repair rates, also positively correlate with T cell number and are significantly suppressed in non-survivors. Simulations show that the lack of T cells can lead to more significant inflammation, proposing an explanation for the monotonic increase of COVID-19 mortality with age and higher mortality for males. We propose that T cells play a crucial role in the immunity against COVID-19, which provides a new direction–improvement of T cell number for advancing current prevention and treatment.

# 1. Introduction

Identifying key factors determining the immune heterogeneity from non-survivors to survivors is crucial for the current fight against the COVID-19 pandemic. Past clinical studies have found a series of host factors associated with severe disease or higher mortality via correlation analysis: individual characteristics including older age, male sex and comorbidities [1–3]; profound lymphopenia, with T cells most significantly affected [4–6]; the elevated level of inflammation markers, like LDH (lactate dehydrogenase) and D-dimer [3,7]; excessive release of pro-inflammatory signalling molecules, like $IFN - \gamma$, IL-6, etc. known as the cytokine storm which is thought likely to be a major cause of multiorgan failure [5,8]. For immune responses, both SARS-CoV-2 specific T cells and antibodies are observed in COVID-19 patients [7,9]. However, the quantitative role of these factors in antiviral and anti-inflammatory immune responses is unknown, resulting in several unsolved questions about the cause of death and the protective mechanism against virus and inflammation: 1. Are T cells and antibodies helpful or harmful [10,11], especially in severe patients? What are the relative contributions of T cell and antibody response for antiviral immunity at different stages? 2. What are the main drivers and suppressors for the cytokine storm and multiorgan failure? Most importantly, 3. Are there new directions to overcome the heterogeneity of patients, decay of antibody function, and gene mutation SARS-CoV-2 in efficient therapeutics and vaccine developments?

Beyond correlative analyses, quantitative modelling is a powerful tool to simulate the measured dynamical immune response to reveal the relative importance of different components [12]. However, many recent studies focus on the dynamics of the virus and its interactions with immune responses [13,14] and antiviral drugs [15–24], without considerations of inflammation which is essential in disease progression. On the other hand, some multiscale simulations [25–28] incorporate existing knowledge about the viral dynamics and immune responses (with inflammation) to simulate the clinical outcomes. However, these approaches involve hundreds of model parameters, which have considerable value uncertainties that limit the reliability of predictions and systematic comparisons with clinical data. Therefore, previous studies either include no inflammation or include too many cellular or molecular inflammation components and parameters to clarify the key factors dominating death.

In this work, we adopt a top-down modelling approach to construct a simple and verifiable model including both antiviral dynamics and inflammation. The model quantifies crucial functional aspects in the virus-immune system battle to overcome the difficulty mentioned above. Here, the battle is classified into three kinds of functional behaviours, namely, the pathogenic function (e.g. virus and inflammation), the protective function (e.g. innate and adaptive immunity) and organ damage. Integrating with the existing clinical and immunological knowledge for COVID-19 patients, we establish a dynamical motif for a small set of crucial functional variables and their interplays. The antiviral inflammation model is used to simulate the systematic progression of COVID-19 patients with 19 parameters that can all be estimated from clinical data. These simulations are validated by real-time clinical data involving hundreds of patients and allow evaluation of contributions of T cells and antibodies to antiviral immune responses. Subsequently, we quantify the difference of anti-inflammatory immune responses from non-survivors to survivors and clarify their correlations to T cells. Finally, T cells' critical role in preventing deaths from COVID-19 and its inspiration to therapeutics and vaccine development are discussed.

# 2. Causal network and mathematic model of the antiviral-inflammation responses

The difficulty of previous multiscale simulations [25–27] due to considerable parameter value uncertainties stems from the fact that, in the bottom-up strategy, the immune response to infectious disease is modelled as a complex network of numerous factors, resulting in the so-called 'curse of dimensionality' [29]. By contrast, in a recent successful model of a classical complex system, namely, fluid turbulence, one of us has demonstrated that the global motions composed of numerous components typically display a symmetry-breaking which can be quantitatively modelled with finite functional variables, called order functions [30,31]. The fundamental distinction of the reference [30,31] from the traditional virus dynamics model is that it postulates the existence of key macroscopic degrees of freedom (called order parameters) of any complex system for each relevant macroscopic function. In many cases, the existence of order parameters is not obvious, but if they do exist (under appropriate statistical average), then parameters in the model are macroscopic on the scale of the

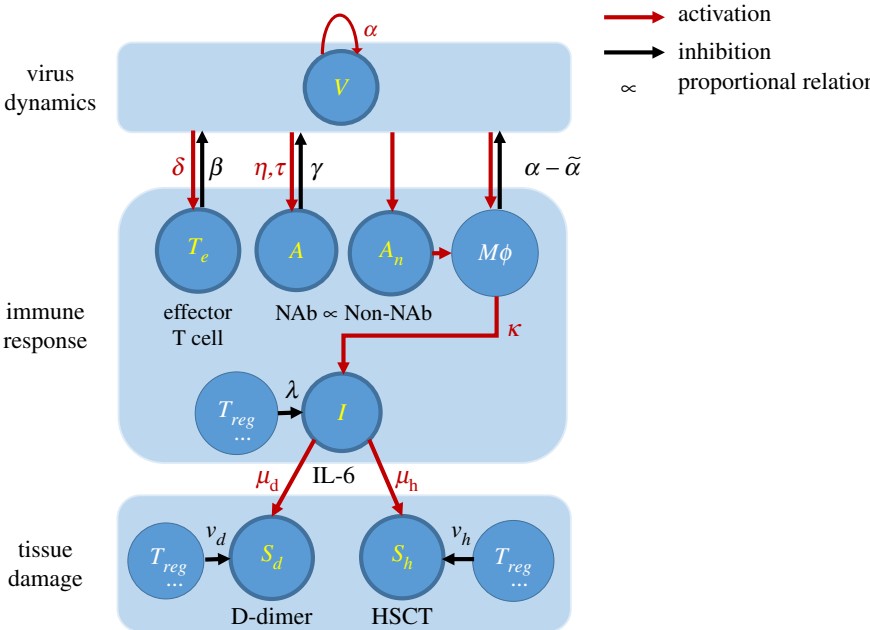

**Figure 1.** Simplified schematic diagram of COVID-19. Key components modelled in this work are highlighted in yellow, i.e. the virus ($V$), effective T-cell ($T_e$), neutralizing antibody (NAb, $A$), non-neutralizing antibody (Non-NAb, $A_n$), interleukin 6 (IL-6), the coagulation marker (D-dimer, $S_d$), the heart injury marker (High-sensitivity cardiac troponin I, HSCT, $S_h$). Red arrows represent activation, and the black arrows represent inhibition. Greek letters are the activation/inhibition rates or characteristic time associated with each interaction. Other components are macrophage (Mφ) and regulatory T cells (Treg), which contribute to the dynamics of the key components but are not explicitly modelled.

human body and days in time, decoupled from complex microscopic molecular processes. Specifically, at the human-body level, we first decompose the system into three functional classes: the pathogenic function, the protective function and the organ damage. Then, we specify dominant components at the cellular or molecular level for each functional class, ignoring other components. Therefore, this approach captures, by intuition, some essential features to observed physiological behaviours and has avoided unnecessary complexity, which results in a curse of dimensionality with little clinical meaning.

The model explicitly describes dynamics of five crucial functional quantities that determine COVID-19 progression: virus ($V$) and interleukin 6 (IL-6, $I$) for pathogenic function, effector T cells ($T_e$) and neutralizing antibodies (NAbs, $A$) for protective function, D-dimer (coagulation marker, $S_d$) and high-sensitivity cardiac troponin I (HSCT, heart injury marker, $S_h$) as examples for multiorgan damage. The six variables, $V$, $T_e$, $A$, $I$, $S_d$ and $S_h$ represent the overall concentrations of the virus, effector T cells, neutralizing antibodies, interleukin 6, D-dimer and high-sensitivity cardiac troponin I of a human body. In figure 1, we classify these six variables and their mutual interactions into viral dynamics, immune response and tissue damage. Furthermore, we quantify the dynamics of these six variables by equations (2.1)–(2.5), which takes the impacts of other physiological factors (such as B cells, macrophages and regulatory T cells) into account in the activation/inhibition rates.

The virus-immunodynamics of COVID-19 has multiple stages during the whole course of infection. Specifically, we propose, it can be divided into three stages in which the model focuses on the second one. The first stage is the early stage of infection when the adaptive response has not been produced, and only the physical barrier (skin, mucosa, cilium) and innate response are actively protecting the body [1]. At the second stage, the adaptive response produces virus-specific T cells and antibodies, which interact against the virus [32]. The third stage is the dying stage of non-survivors; the virus expands to the whole body [33], the target cells may be depleted, and protective immune pathways are massively blocked [8]. We believe the success or failure of the virus-host battle during the second stage determines whether the patients can survive or not, so we focus on the second stage [34]. We assume that, in this stage, the viral load is growing exponentially due to sufficient target cells, and virus-specific T cells and antibodies are produced significantly to suppress the virus. Therefore, we neglect the slight initial delay of the effector T cell production (i.e. the activation of naive T cells) and the ultimate target cell limitation.

Specifically, for the production of effector T cells, initially naïve CD8+ T cells are activated by the free virus through antigen presentation by macrophages, dendritic cells and CD4+ cells, or directly by

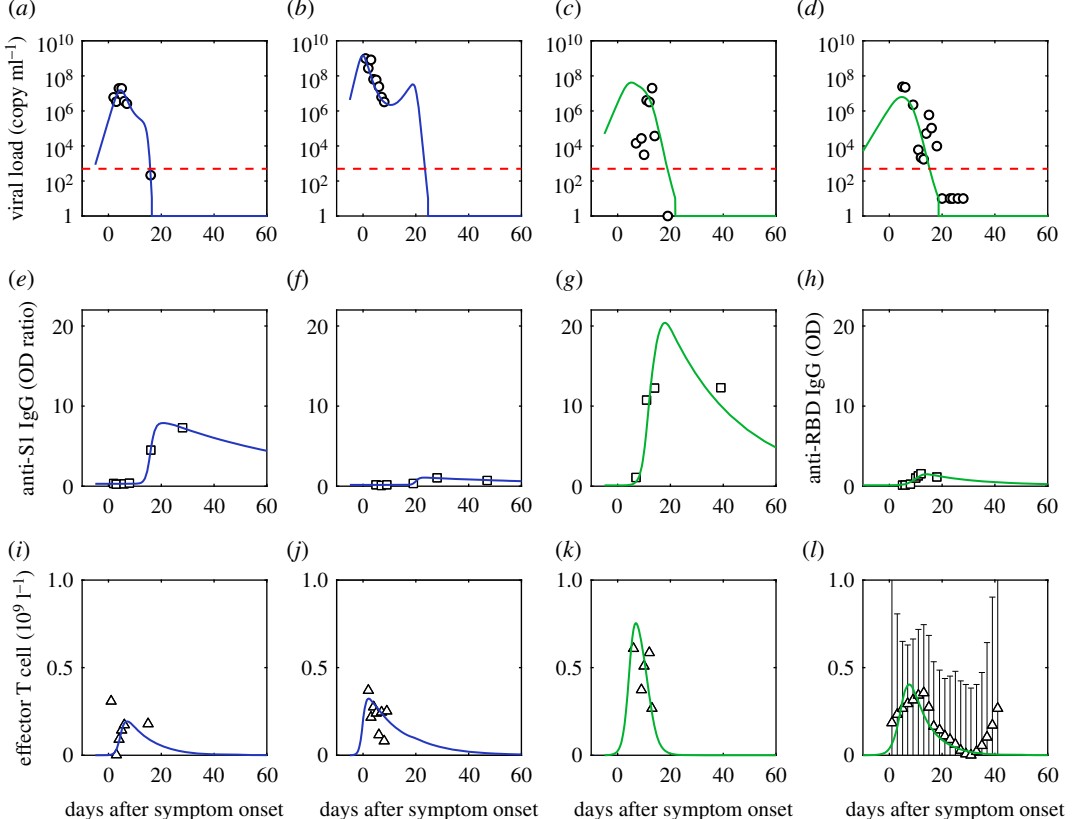

**Figure 2.** Comparison of simulation of viral dynamics and adaptive immune response to data. (*a,e,i*) are fits of the first patient (patient ID labelled as P1); (*b,f,j*) are fits of the second patient (patient ID labelled as P3); (*c,g,k*) are fits of the third patient (patient ID labelled as P5) [35]; (*d,h,l*) are fits of the fourth patient (patient ID labelled as 902). Mild patients are in blue and severe patients are in green. Red dashed lines are the limit of detection [36]. Black dotted lines are normal ranges [4]. Viral load is from the nasopharyngeal swab. The effective T cell data is the reduction of CD3+ T cells in serum ($T_{serum}$) from its initial value ($T_0$), which is assumed proportional to the concentration of effector T cells in organs. We assume the CD3+ T cell data of the patient with the same severity share similarities based on the distinction of CD3+ T cell data between groups of different severities [4] and approximate the CD3+ T cell dynamics of the fourth severe patient using the median data of the severe group [4].

infected cells [7]. Naïve CD8+ cells then differentiate and go through clonal expansion into effector T cells [10]. We describe this production kinetics with $\delta V(t)$ in equation (2.2), where $\delta$ is the integrated rate of the whole pathway from virus to effector T cells. The maximum variation of the T cell concentration is observed to be close to the viral load peak location (e.g. figure 2*j,k*), revealing a slight and negligible delay time for T cell production. Following activation, most effector T cells undergo apoptosis [37], which is described by a simple decay term, i.e. $-\epsilon T_e(t)$.

For the production of neutralizing antibodies, initially, naive B cells are activated directly by the free virus or indirectly through antigen presentation by macrophages, dendritic cells and CD4+ cells. Naive B cells then proliferate and differentiate into plasma cells to synthesize and secrete neutralizing antibodies [7], which are gradually degraded in serum [38]. In contrast to the quick production of effective T cells, the antibody production is described by delayed kinetics as $\eta V(t - \tau)$ in equation (2.3), where $\tau$ represents the time delay caused by the entire pathway from the entry of the virus into the body to the antibody secretion, and $\eta$ is the integrated production rate in this pathway. This delay kinetics is consistent with clinical evidence that the antibody secretion is observed on the 5–15th day after onset [36,38,39], later even than the date of viral load peak. Therefore, antibody response includes a delay, while equation (2.2) for T cell response does not. On the other hand, we assume a similar simple decay for antibody, characterized by $-\theta A(t)$ in equation (2.3).

For viral concentration, the free virus enters the target cell and begins replication, thus increasing the viral concentration, while the phagocytes engulf the free virus and infected cells (innate immunity), thus decreasing the viral concentration [34]. We use $\tilde{\alpha}V(t)$ in equation (2.1) to represent the effective growth rate of the virus incorporating the above two processes. Because of the fact that in the early stage, the

virus increases exponentially at a single rate [14,16,17,22–24]; and in the later stage, T cell and antibody response significantly [4], the innate immunity is unable to control the virus; thus, the variation of virus clearance rate from the innate response is negligible, and therefore, $\tilde{\alpha}$ is set to be constant. For adaptive immunity, the effective T cells kill the infected cells, while the neutralizing antibodies bind to the free virus, preventing the virus from entering the cells, and the neutralized virus is cleared by other macrophages [34]. These two antiviral-immune behaviours both reduce the concentration of the total virus and are described by the common [40,41] bilinear dynamics $-\beta T_{e}(t)\,V(t)$ and $-\gamma A(t)V(t)$ with integrated rate parameters $\beta$ and $\gamma$, where we assume the concentration of infected cells and the free virus is proportional to the total virus concentration, $V(t)$.

On the other hand, B cells secrete neutralizing antibodies accompanied by non-neutralizing antibodies [35], which is very important for IL-6 dynamics. Specifically, it is suggested [42] that non-neutralizing antibodies bind to macrophages and promote macrophages susceptible to the virus, and accompanied with this process, macrophages secrete a large amount of IL-6. Based on this, we propose that IL-6 production by non-neutralizing antibodies dominates the cytokine storm taking place at the later stage [3] and neglect the IL-6 produced by T cells, monocytes and infected cells [34]. Therefore, we assumed that the production rate of IL-6 is proportional to the concentration of non-neutralizing antibodies, which is assumed proportional to the concentration of neutralizing antibodies. In this context, we use $\kappa A(t)$ in equation (2.4) to describe the production of IL-6, where $\kappa$ quantifies the integrated impacts of the whole pathway from non-neutralizing antibodies to IL-6 production. Meanwhile, we use $-\lambda[I(t) - I_0]$ to describe the negative feedback mechanism of IL-6, where $I_0$ is the normal value.

For tissue damage, the cytokine storm recruits a large number of dysregulated immune cells that cause damage to organs such as the heart and blood vessels [8,34]. Specifically, damaged cells in the heart release HSCT and coagulopathy leads to D-dimer production [8]. On the other hand, the regulatory T cell and other T lymphocyte populations can secrete pro-repair factors, modulate repair in the injury environment [43], and thus reduce the level of tissue damage markers. Therefore, we use $\mu_d I(t)$ and $\mu_h I(t)$ in equation (2.5a) and (2.5b) to describe the increasing dynamics of D-dimer and HSCT, where $\mu_d$ and $\mu_h$ is the integrated rate characterizing the impacts of the entire pathway above from IL-6 to D-dimer and HSCT. We use $v_d S_d(t)$ and $v_h S_h(t)$ to describe the curing effect. The detailed units of all parameters are summarized in electronic supplementary material, table S1.

$$\frac{dV(t)}{dt} = [\tilde{\alpha} - \beta T_{e}(t) \, - \gamma A(t)]V(t), \tag{2.1}$$

$$\frac{dT_{e}(t)}{dt} = \delta V(t) - \epsilon T_{e}(t) \,, \tag{2.2}$$

$$\frac{dA(t)}{dt} = \eta V(t - \tau) - \theta A(t), \tag{2.3}$$

$$\frac{dI(t)}{dt} = \kappa A(t) - \lambda[I(t) - I_0], \tag{2.4}$$

$$\frac{dS_d(t)}{dt} = \mu_d I(t) - v_d S_d(t) \tag{2.5a}$$

and
$$\frac{dS_h(t)}{dt} = \mu_h I(t) - v_h S_h(t) \,. \tag{2.5b}$$

Simulations of equations (2.1)–(2.5) are compared to real-time data with 457 patients involved with 10 individuals. To describe the coupled evolutions of the virus, effector T cells, antibodies, IL-6, D-dimer and HSCT for a single patient or group, we have to use a complete set of data for all these six quantities; however, there is no literature that reports such a complete set in one time. Therefore, we use data from multiple sources to construct the complete data set. Each data set is constructed from data sources of the same severity, and there are in total four data sets: mild, severe, survivor and non-survivor group. For detailed data sources and integration of data from different sources, see Methods and electronic supplementary material, table S2.

For individuals, we fit each patient's data directly by equations (2.1)–(2.5). While for groups, we fit the median values of each variable by equations (2.1)–(2.5) for we have shown that the group dynamics of patients satisfy a similar set of equations with ensemble-averaged parameters (see electronic supplementary material). We perform the least-square fit of data using the *fmincon* function of MATLAB with the implemented interior-point optimization algorithm and perform numerical simulation using the delayed differential equations (DDE). The objective function for equations (2.1)–(2.5) are shown by equations (5.5)–(5.7) in Methods. The time axis for simulation is the number of days after symptom onset, with the starting day, $t_0$ being 0 or several days earlier. Besides, according to equation (2.2), the

production rate of effector T cells is the maximum when the virus peaks and is several magnitudes smaller on $t_0$ when the virus is low. Therefore, we approximate the initial effector T cell and antibody concentration to zero because there is no virus-specific effective T cell or antibody before infection [44]. For a detailed description of the fitting procedure, see Methods and electronic supplementary material.

# 3. Results

## 3.1. Viral dynamics and contributions of T cells and antibodies to antiviral responses

Virus, effector T cell and antibody dynamics are simulated to compare with real-time data from 10 individuals [35,36] and median values of mild and severe (critical) groups, survivors and non-survivors [3,45,46]. The concentrations of virus ($V$) and antibodies ($A$) are assumed proportional to viral load measurement from the respiratory tract and optical density or titre of Anti-RBD IgG/Anti-S1 IgG/Anti-NP IgG. For T cell data, due to the fact that it is difficult to obtain systematic data (data that have continuity along time, broad severity spectrum and a large number of patients) of effector T cells, we use CD3+ T data in serum from Zhang *et al.* [4]. Because it is difficult to detect the effective T cell number inside organs, we estimate the effector T cell number from the experimentally measured CD3+ T cell data in serum. Zhang *et al.* [4] deduced that the reduction of CD3+ T cells in serum represents the number of T cells that entered organs and produced effector T cells. In this context, we define, at the present stage, the amount of effector T cells ($T_e$) is proportional to the reduction of CD3+ T cells in serum ($T_{serum}$) from its initial value ($T_0$). In figure 2a–c, the data of $T_e$ come from [35], while in 2d, they come from [4] and the $T_{serum}$ count is mapped from the original lymphocyte count scaled by 0.589 (Methods). Besides, when comparing simulation to antibody data, an instrumental baseline value $B_0$ (usually observed in the experiment) [36,44,45] is added to $A(t)$.

As shown in figure 2 and electronic supplementary material, figure S1, simulations show agreement to data for all cases; for parameters, parameter uncertainties and the goodness of fit, see electronic supplementary material. Though the fit is done separately for each individual/group with different initial guesses due to the limited number of data points and large fluctuations, the parameters for different patients and groups are within the same order of magnitude (for example, $\tilde{\alpha}$ varies from 0.41 to 1.86). The goodness of fit is defined as 1 - root mean square/maximum of the simulation of the variable. The mean of the goodness of fit among all variables and patients is $91.8\% \pm 6.6\%$. See electronic supplementary material, tables S8 and S9 for parameters and the goodness of fits of each variable in each case. This stability implies that the fitting approach is credible.

We choose to model a dynamic with the basic time scale of days. In this modelling, the doubling time of the viral load equals to $\ln 2/\tilde{\alpha} = 1.0 \pm 0.6$ days, meaning the virus takes $1.0 \pm 0.6$ days to double the amount by replication, which is consistent with the slope of viral load (log scale) in the increasing stage (fig. 4 in [40]). The effector T cells have a half-life equal to $\ln 2/\epsilon = 11.2 \pm 9.8$ days, consistent with the general observation that most effector T cells undergo apoptosis following viral clearance [37]. The antibodies have a half-life equal to $\ln 2/\theta = 13 - 57$ days (decline by 50%), which is also consistent with the recent observation of longitudinal decline of neutralizing antibodies by more than 90% after 60–100 days post onset [38].

For effector T cells, there is early dynamics for activation before any measurements. Then using our model, we can predict the early T cell behaviours by fitting the T cell data after onset, as shown in figure 2j and electronic supplementary material, figure S2a (3rd line). We suggest further measurements of T cells in latent patients to verify the proposed effector T cell activation dynamics.

Clarifying deterministic factors controlling viral load peak benefits early antiviral treatment, vaccination and epidemiological control [47]. To understand the main factors that determine viral peak, by asymptotic analysis (Methods), we get an analytical solution that predicts the peak is determined from virus inhibition by T cells: $\tilde{\alpha}^2/2\beta\delta$. This gives $10^{7.14}$ copy/ml for the viral peak of patient P1 in figure 2 consistent with the simulated value $10^{7.18}$ copy/ml. Electronic supplementary material, table S3 summarizes the analytical viral peak values compared to the corresponding simulation, which have >90% overlap with each other for all cases. Data agreement gives that for patients who survive, on average, 95% of viruses are cleared per day with $10^9/l$ CD3+ T cells from blood engaged, revealing strong efficiency of T cells' virus clearance. The consistency between simulation and data supports our hypothesis about the antiviral dynamics in which adaptive response plays a significant role—first,

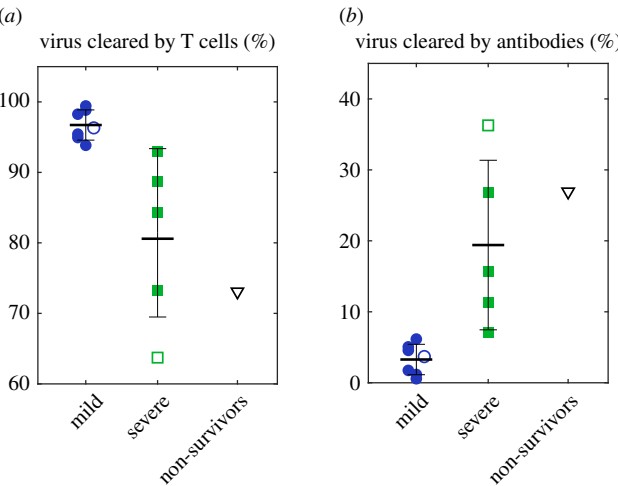

**Figure 3.** An overall statistic of the fraction of virus killed by T cells (*a*) and antibodies (*b*) for all cases. Solid markers are individual data, and hollow markers are group data. Error bars represent standard errors.

effector T cells are activated, kill infected cells, and induce viral peak; then neutralizing antibodies are secreted, and finally, clear the virus.

Equation (2.1) has simplified the antiviral responses of adaptive immune processes, i.e. effector T cells kill infected cells, and antibodies inactivate the virus particles, to the common bilinear kinetics [40,41]. One advantage of this simplification is that it enables us to quantify (although preliminarily) the roles of T cells and antibodies in the antiviral process. In this context, we define the amount of virus cleared by T cells, $N_T$ and antibodies, $N_A$, are: $N_T(t) = \int_{t_0}^{t} \beta T(t') V(t')\, dt'$ and $N_A(t) = \int_{t_0}^{t} \gamma [A(t') - A_0] V(t')\, dt'$. Then, the contribution by T cells in adaptive response for clearing virus, $F_T$, and the contribution of antibodies, $F_A$, are: $F_T = \int N_T(t)\, dt / [\int [N_T(t) + N_A(t)]\, dt]$, $F_A = 1 - F_T$.

For patients of different severities, we compare the quantitative contributions of T cells and neutralizing antibodies for virus clearance, as displayed in figure 3. It shows T-cell immunity dominates the total virus clearance for all patients (88.8%) but significantly decreases from mild to severe patients, consistent with previously reported less CD4+, CD8+ response in severe patients compared to mild patients [9]. Instead, the antibodies' contributions are 3.3% (mild), 19.4% (severe or critical) and 28.9% (non-survivors), respectively. Our simulation finds that the antibody preparation time before secretion is smaller overall in severe cases (9.42, 5.40–12.79 day) than in mild cases (14.68, 8.45–20.13 day), revealing antibodies in severe patients secrete earlier and cleared more virus. This finding reveals an important inference in COVID-19 that T cells may have a dominant role in the virus clearance relative to antibodies, especially for mild patients.

## 3.2. Inflammation dynamics associated with death

To clarify the main driver for the cytokine storm and organ damage of critical illness, we compare simulations of equations (2.1)–(2.5) with real-time, median data of survivors and non-survivors (figure 4). The concentration of non-neutralizing antibodies is assumed proportional to anti-RBD IgG optical density. For data source and parameter estimation, see Methods and the electronic supplementary material. Although the observed IgG data of survivors and non-survivors solely are incomplete to estimate the saturation value of antibodies, based on a complete measurement of critical and non-critical patients [48], we use the ratio of the saturation values of critical and non-critical patients to approximate the ratio of the saturation values of non-survivors and survivors, which guides the simulation of IgG in figure 4. The agreement between the experiment and the data supports the validity of equations (2.1)–(2.5) and allows us to investigate the critical difference between survivors and non-survivors.

IL-6 formation rates are assumed to be the same for both groups. The striking feature of non-survivors compared to survivors is the continuous production of IL-6 and organ damage, revealed by zero inhibition rates for all three markers (figure 5*b* where the three rates are overlapped), while the difference of formation rates of organ damage markers is not remarkable. The zero-inhibition rate of IL-6 for non-survivors may also cause more uncontrolled inflammation before symptom onset so that non-survivors have higher IL-6 than survivors at the 0th day after onset (figure 4*d*). Our finding implies that the crucial aspect of death from COVID-19 is the lack of negative feedback for anti-inflammatory cytokine inhibition and tissue repair.

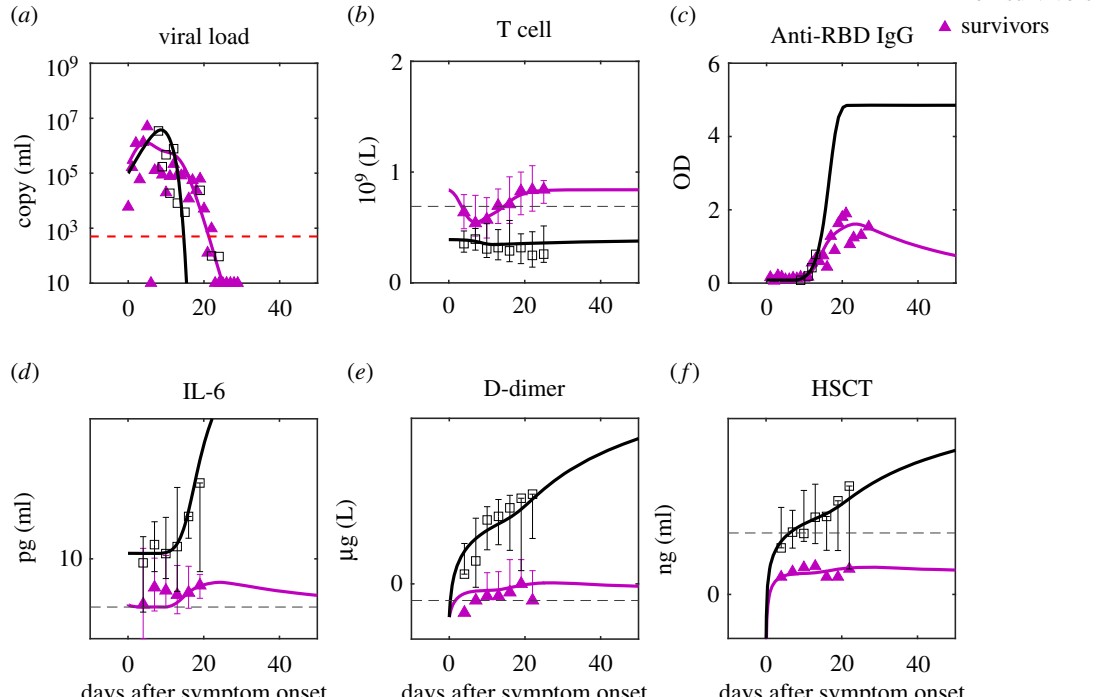



**Figure 4.** Comparison of predictions to clinical data of survivors and non-survivors. The data are the median of the group with error bars [3]. For parameter estimation, see Methods and electronic supplementary material. For IgG simulation, we use the ratio of the saturation values of critical and non-critical patients [48] to approximate the ratio of the saturation values of non-survivors and survivors. The red dashed line is the limit of detection and the black dashed lines are the normal ranges of the corresponding markers (see the reference from the electronic supplementary material).

## 3.3. Initial T cells as background immunity that reduce mortality

Our model suggests that virus clearance by T cells, cytokine inhibition and tissue repair are three essential protective functions in COVID-19 that determine disease severity. To seek what determines these protective functions, T cells' contribution of total virus clearance, cytokine inhibition rate and tissue repair rates are plotted with initial T cell concentration before infection (equal to T cell baseline, $T_0$) in figure 5a,b, which shows a positive correlation (figure 5b is based on statistics of 137 non-survivors and 54 survivors). It shows the importance of sufficient initial T cells for comprehensive protection, which comes from an adequate number of effective CD8+ effector T cells that kill infected cells, sufficient regulatory T cells and other subsets that suppress the immune response and promote tissue repair [43] to reduce over inflammation in non-survivors.

A great public concern is how an individual patient's background 'immune health' landscape (simplified as background immunity) shapes responses to SARS-CoV-2 infection [10] and controls the disease's severity. Because of the positive correlation found between protective functions and initial T cell concentration, T cells' static reserve before infection and dominance in population compared to other cells against the virus, we hypothesize that concentration of initial T cells is a crucial characterization for the background immunity against SARS-CoV-2. To justify this hypothesis, we conduct disease progression of patients with different initial T cell concentrations (Methods). According to figure 5a,b, by assuming a linear decrease of T cells' virus clearing rate, IL-6 and D-dimer inhibition rates with decreasing initial T cell concentration, figure 4c shows the coagulation becomes more and more significant, which means the lack of initial T cells exacerbates disease severity and increases mortality risk.

Therefore, we propose that the T cells' impaired antiviral and anti-inflammation functions are the main immune origin of death from COVID-19: the extremely low level of initial T cells in non-survivors results in weak antiviral, cytokine inhibition and tissue repair abilities; then it calls the elevated antibodies for compensation; as a result, the concomitant large amount of non-neutralizing antibodies amplifies the cytokine storm, leading to continued damage. Following this casual chain, according to the decrease of lymphocytes (hence decrease of T cells assuming T cell count proportional to lymphocyte count with a

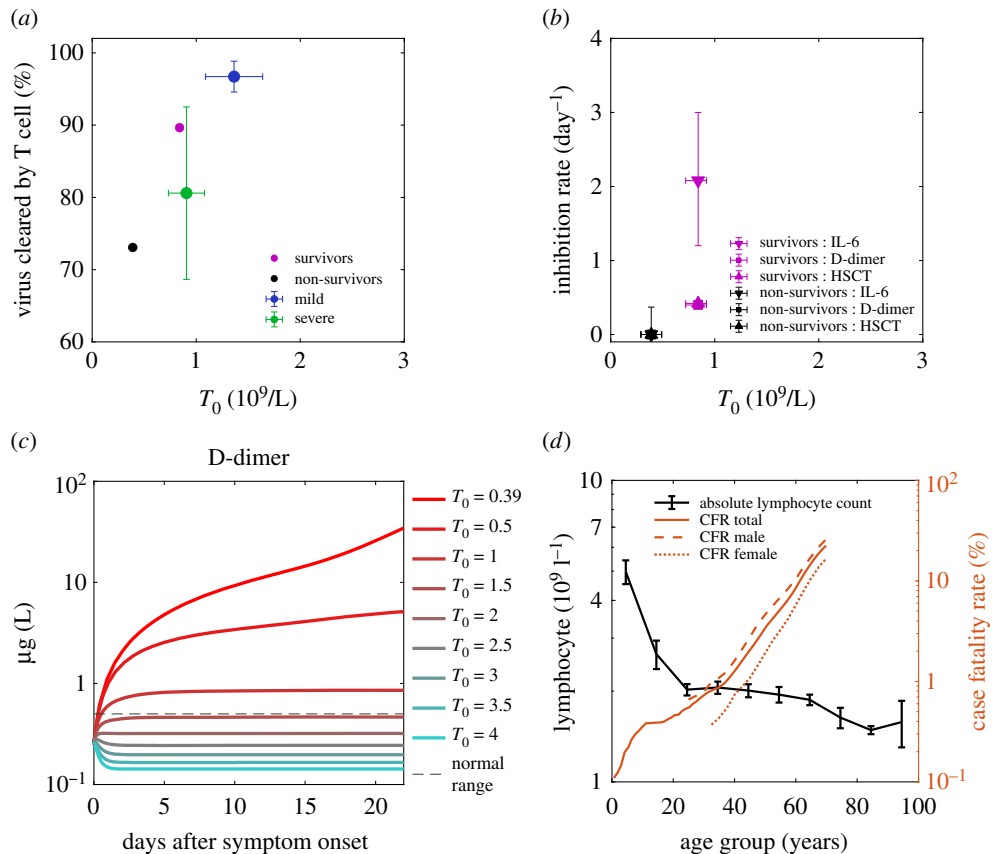

**Figure 5.** Initial T cell concentration as the background immunity of individuals against SARS-CoV-2 and reduces mortality. (*a*) Positive correlation of T cell's antiviral contribution with initial T cell concentration for mild patients (blue), severe patients (green), survivor group (magenta) and non-survivor group (black). Data points are the means, and error bars are the standard deviations. (*b*) Positive correlation of inhibition rates of IL-6, D-dimer and high sensitive cardiac troponin (HSCT) with initial T cell concentration for survivor group and non-survivor group. Data points are the means, and error bars are the standard deviations. It is worth mentioning that the three rates of the non-survivor group are all zero and overlap with each other; The inhibition rates of D-dimer and HSCT of the survivor group are closed and overlap with each other. (*c*) D-dimer dynamics of non-survivors with an increase of initial T cell concentration ($T_0$) reduces organ damage at the later stage. The Red dashed line is the normal upper limit of the D-dimer [4]. For parameters of the simulation, see Methods. (*d*) Lymphocyte count decreases with age, and mortality (case fatality rate, CFR) increases with age. Males (dashed line) have higher mortality than females (dotted line).

coefficient 0.589, see Methods) with older age (figure 5*d*), we anticipate straightforwardly that older patients must have higher mortality than younger patients. Also, male patients should have higher mortality than female patients for their lower CD4+ T cells[37]. This inference proposes an explanation for the continuous increase of COVID-19 mortality with age and higher mortality for males [49], shown in figure 4*d*. Another hypothesis, for example, one states that men produce lower levels of antibodies than women [50]. According to our model, T cells are expected to have higher impacts on virus clearance than antibodies; therefore, lower levels of antibodies are less likely to cause worse protection against the virus for men than lower levels of T cells. We propose to measure dynamics of effector T cells, antibodies and viral load in women and men and calculate the fractions of virus cleared by T cells ($F_T$) and antibodies ($F_A$) using the model. If antibodies are less protective in men as the other hypothesis states, we expected to see a lower $F_A$ and higher $F_T$ in men than women. If it is the T cells that are less protective in men as suggested by our model, we expect to see a lower $F_T$ and higher $F_A$ in men than women.

## 4. Discussion

COVID-19 is a complex disease that incorporates viral dynamics, multiple immune responses and organ damage together, with strong clinical and public needs. Therefore, we developed a simple, easily

applicable model which covers all three aspects. It aims at revealing macroscopic interactions among the entire amount of the virus and host factors and discover the critical factor for survival and provide insights for clinical application. In this work, we have quantified the adaptive-immune-response heterogeneity from non-survivors to survivors of COVID-19, using a dynamical motif with 19 measurable parameters, which complements the complicated multiscale model [27]. First of all, this model provides an accurate description of real-time clinical data involving hundreds of patients, which then support T cells' critical roles in the antiviral and anti-inflammatory immune responses of COVID-19. Furthermore, beyond the previous correlation analysis for T cell scarcity and disease severity [9,10], a causal relationship between death from COVID-19 and impaired T cell immunity is supported by this work, providing a preliminary explanation for the high mortality of older men.

It is worth mentioning that these important findings are inferences based on our assumptions of immunodynamics (i.e. equations (2.1)–(2.5)) and are preliminarily supported by the present work. Therefore, considering the strong clinical needs, there is an urgent call to investigate them in more detail. Equations (2.1)–(2.5), the associated assumptions and the critical role of T cells need to be tested by more real-time data. For instance, we suggest further studies on two aspects. One is to verify whether the T cells play a dominant role in adaptive immunity. On the one hand, equations (2.1)–(2.3) should be applied to more clinical data of individual patients, which should be measured for viral load, CD8+ and neutralizing antibodies simultaneously. On the other hand, it is important to compare the effectiveness of vaccines or drugs that activate (or suppress) antibody and T cell immunity, respectively. The other is to verify whether the production of IL-6 is dominated by the combination of non-neutralizing antibodies and macrophages. We propose to measure non-neutralizing antibodies, macrophages and IL-6 along time in clinical patients and then quantify their real-time correlations. We anticipate, at the early stage of the cytokine storm, the secretion rate of IL-6 would be proportional to the product of macrophage and non-neutralizing antibody concentrations, i.e. the first term on the right-hand side of equation (2.4). This kind of verification can be applied to other assumptions as well, such as whether viral clearance rate from the innate response is constant around symptom onset and whether effector T cell concentration is proportional to the reduction of CD3+ T cell concentration in peripheral blood.

According to our discussion (figure 4*a*,*b*) of mild and severe patients, survivors and non-survivors, a better vaccine or treatment requires better protective functions, i.e. higher virus clearance rate, cytokine inhibition rate and tissue repair rate, either before or during the infection. As suggested by figure 4*a–c*, prevention and treatment approaches that improve (active) T cell number before or during the infection have the potential to enhance the three protective functions together and thus give higher efficacy.

The currently tested drugs target various pathogenesis levels, from antiviral to anti-inflammatory drugs and antithrombotic agents [27], etc. One crucial challenge is the lack of broad applicability of these drugs to heterogeneous patients with various comorbidities, disease severities and complications [51]. Our study provides a promising direction—increasing T cell number and functions by both drugs and health care activities, which may benefit virus clearance, cytokine inhibition and tissue repair simultaneously. For instance, there is some recent evidence showing Chinese herbal medicine can improve the number and function of different T cell subsets, like CD4+ and CD8+, see recent review by Robert D. Hoffman and reference therein. Specifically, for COVID-19 patients, it is been found herbal medicine can obviously improve lymphocytes and shows a remarkable therapeutic effect, like Shufen JieDu [52,53], and others reported from clinical treatment [54]. Furthermore, the mouse model found Shufen JieDu improves CD4+ and CD8+, significantly reducing the virus load in the lung from 1109.29 ± 696.75 to 0 ± 0 copies/ml, and reduces the cytokine level [52]. Similarly, TaiChiQuan and meditation have been found to have the effect of increasing CD4+ T cells from several reports, respectively [55–57]. Also, for the recovery of COVID-19 patients and healthy people's prevention, TaiChiQuan and meditation can potentially benefit a wide range of ages, including older adults. Therefore, we strongly suggest studying their curing and immunity improvement effects against COVID-19.

For clinical application, to maximize the curing effect for severe patients, we suggest adopting multistage, synthetic protocols incorporating the above therapies. In this case, our model provides a useful tool to evaluate the effectiveness of treatments to identify individual optimal protocols. The reason is that all parameters can be determined from clinical data and quickly predict individual patients' trajectories, which may also advance the early prediction algorithm of current artificial intelligence softwares [58,59].

For COVID-19 vaccination, the protective functions of T cells presented in this study support the future vaccine development targeting stimulation of T cells. Currently, there are several widely used vaccines, e.g. BNT162b2 (Pfizer–BioNTech, two-dose), AZD1222 (Astra Zeneca–University of Oxford,

two-dose), Ad26.COV2-S (Johnson & Johnson, one-dose). The reported impact of these vaccine efficacies on SARS-CoV-2 variants are: BNT162b2 showed 90% efficacy for Alpha and 75% for Beta; AZD1222 showed 75% on Alpha and 10% for Beta; Ad26.COV2.S shows 70% for Alpha, and for Beta, it showed 72% efficacy in the USA, 66% in Latin America and 57% in South Africa. Especially, Ad26.COV2-S showed 85% effectiveness in preventing severe cases across the USA, Latin America and South Africa with one dose vaccine and sustained (and increasing) immune protection over time, which is suspected by former FDA Commissioner Scott Gottlieb, MD to come from a robust early induction of memory T-cells (CD4+ and CD8+) [60]. This is consistent with the model and is supported by the recent observation that T memory cells sustained in convalescent patients for more than ten months [61] and virus-specific T cells are found even in uninfected people [44,62]. Relative to the short-term decay (decline by 90% after 60–100 days post onset [38]) of antibodies, the long-term existence of T-cell memory means the vaccines associated with T cells activation have the advantage of potential long-term protection, so it has become one of the main goals of future vaccine development [61–64].

In the future, the scope of the model can be expanded. In our three-stage perspective, we suspect that the current model may be modified for asymptomatic patients and the dying stage of non-survivors by including significant innate response and effect from target cell limitation. For instance, a temporal evolution of the innate response (accompanied by IL-6 production) may explain the small deviation of the simulation from IL-6 dynamics data of survivors on the 7th–13th day (shown in figure 4*d*). Therefore, our model can be expanded to distinguish the impact from the innate response, T cell response, antibody response and target cell limitation in the future.

# 5. Methods

## 5.1. Extraction of data from published literature

A software tool WebPlotDigitizer (https://automeris.io/WebPlotDigitizer), was used to extract data from fig. 2 in ref. [4], figs. 1 and 3 in [35], fig. 2 in [3], figs. 1 and 3 in [45] and fig. 3 in [46]. All extracted data were made available to readers in our GitHub shared folder: https://github.com/luhaozhang/covid19_openscience.git.

## 5.2. Data source and integration of data from different sources

There are in total ten individuals and four groups in this work. Four patients are from Isabella Eckerle's cohort [35] and six patients from Kelvin To's cohort [36], and the severity classification follows the assignments in previous publications. One patient from Kelvin To's cohort has not been identified as severe or critical. Because in the original cohort, it has a low probability to be critical, in this paper, we assign it as severe. In all, there are six mild patients and four severe (including critical) patients. The four groups are mild, severe (including critical), survivors and non-survivors.

The viral load data are from the oropharyngeal swab/posterior oropharyngeal sample/endotracheal aspirates/nasopharyngeal swab sample measured during the 0th to 30th day after onset. T cell, Lymphocyte, antibody, IL-6, D-dimer and HSCT are measured from serum sample during the 0th to 30th day after onset. Lymphocyte data were multiplied by 0.589 to estimate T cell concentration (0.589 is the ratio between medians of normal ranges of T cells [4] and lymphocytes [35]). See electronic supplementary material, table S2 for how the virus, T cell and antibody data for all individuals and groups are integrated from various data sources. The IL-6, D-dimer and HSCT data for survivors and non-survivors are all from BinCao's cohort [3].

## 5.3. Least-square fit of virus, immune response and inflammation data

For parameters of simulations in figures 2 and 4, we adopt a best-fit approach to find the parameters which minimize the given objective function: the mean of residual sum of squares (RSM) between data points and the corresponding model simulations as used similarly in influenza model [65]. For virus-T cell-antibody dynamics, the objective function is:

$$\overline{\text{RSM}} = \frac{1}{n_V} \sum_{i=1}^{n_V} \left[ \frac{(\log_{10} V_i - \log_{10} \bar{V}_i)}{\log_{10} V_{\max}} \right]^2 + \frac{1}{n_T} \sum_{i=1}^{n_T} \left[ \frac{(T_i - \bar{T}_i)}{T_{\max}} \right]^2 + \frac{1}{n_A} \sum_{i=1}^{n_A} \left[ \frac{(A_i - \bar{A}_i)}{A_{\max}} \right]^2, \qquad (5.1)$$

$V_i$, $T_i$, $A_i$ represent values of viral load data, T cell count data and antibody data, respectively. $\bar{V}_i$, $\bar{T}_i$ and $\bar{A}_i$ represent the corresponding simulated viral, T cell and antibody value given by our model, respectively. $V_{max}$, $T_{max}$ and $A_{max}$ represent the maximum value among viral load data, T cell count data and antibody data, respectively. $n_V$, $n_T$, $n_A$ are the total number of viral load, T cell count and antibody data points used for the parameter optimization. For the objective function of inflammation response, the mean of RSM (equation (5.2)) was used in linear scale for survivors and log scale (equation (5.3)) for non-survivors because the data covers more than one magnitude. $I_i$, $S_{d_i}$, $S_{h_i}$ represent values of IL-6 data, D-dimer data and HSCT (high-sensitivity cardiac troponin I) data, respectively and $\bar{I}_i$, $\overline{S_{di}}$, $\overline{S_{hi}}$ are the corresponding simulated IL-6, D-dimer and HSCT value by our model. $I_{max}$, $S_{d_{max}}$, $S_{h_{max}}$ represent the maximum values among IL-6, D-dimer and HSCT values separately.

$$\overline{\text{RSM}} = \frac{1}{n_I}\sum_{i=1}^{n_I}\left[\frac{(I_i - \bar{I}_i)}{I_{max}}\right]^2 + \frac{1}{n_{S_d}}\sum_{i=1}^{n_{S_d}}\left[\frac{(S_{d_i} - \overline{S_{d_i}})}{S_{d_{max}}}\right]^2 + \frac{1}{n_{S_h}}\sum_{i=1}^{n_A}\left[\frac{(S_{h_i} - \overline{S_{h_i}})}{S_{h_{max}}}\right]^2 \tag{5.2}$$

and

$$\overline{\text{RSM}} = \frac{1}{n_I}\sum_{i=1}^{n_I}\left[\frac{(\log_{10}I_i - \log_{10}\bar{I}_i)}{\log_{10}I_{max}}\right]^2 + \frac{1}{n_{S_d}}\sum_{i=1}^{n_{S_d}}\left[\frac{(\log_{10}S_{d_i} - \log_{10}\overline{S_{d_i}})}{\log_{10}S_{d_{max}}}\right]^2 + \frac{1}{n_{S_h}}\sum_{i=1}^{n_A}\left[\frac{(\log_{10}S_{h_i} - \log_{10}\overline{S_{h_i}})}{\log_{10}S_{h_{max}}}\right]^2. \tag{5.3}$$

For simulation of viral load dynamics, when $V < 10$ copy/ml, it is thought to be cleared thoroughly at one time without further evolution and is set to be one copy/ml. The fit that minimizes the objective function with largely fluctuating data points eliminated is called the best fit, which gives results in figures 2 and 4. Electronic supplementary material, table S5 lists the data points used for performing the best fit of each case. The principles that we use to eliminate largely fluctuating data points are: for virus, data points that represent negative results of the virus are abandoned unless they were important indicators of the ending of viral activity; the data points associated with the second viral load peak in the mild and severe group are abandoned because the phenomenon is not observed as the common feature of individual patients. The decay of CD3+ T cells of 902 and 910 patients after 30 days is not used for fit because of the large 95% CI of the data.

*fmincon* function of MATLAB (MathWorks, version 2012 and higher) with the implemented interior-point optimization algorithm is used to perform the fits. It requires constraints for the parameters to be optimized and an initial guess. An empirical fitting is performed for each patient and group to identify initial guess and parameters' constraints for optimization, shown in electronic supplementary material, table S10 and table S11. Patients with the same type of data and same severity category are set to have similar parameter ranges for optimization. A random initial guess is not suitable here because the fit is sensitive to the initial space, probably because of the limited number of data points with relatively large fluctuation, especially for viral load. For fits of survivors and non-survivors, we first fit virus, T cell and antibody data, then fix relevant parameters and perform fit of IL-6, D-dimer and HSCT data.

Estimation of the uncertainty of parameters is carried out after the best fit for each case. For parameters related to viral dynamics, $\tilde{\alpha}$, $V_0$, $\beta$, $\gamma$, $\delta$, $\eta$, $\tau$, and inflammation-related rates, $\kappa$, $\lambda$, $\mu_h$, $v_h$, $\mu_d$, $v_d$, we use a similar approach to estimate parameter uncertainty as that in Marchingo, J. M. *et al*. Science. 346, 1123–1127 (2014). They use an artificial dataset containing data fluctuation to perform fits to obtain parameter distribution, and we use data containing largely fluctuating points. The parameter values obtained by best fit with no largely fluctuating data are the most probable value. Therefore, the model parameters are identifiable. For other parameters, we use a 95% confidence interval of their non-linear fit or normal physical range. See electronic supplementary material, table S4 for a detailed summary of methods that give the uncertainty for each parameter and electronic supplementary material, table S6 and S7 for best fit and uncertainties of parameters for all cases.

## 5.4. Simulation of D-dimer dynamics with different initial T cell concentration

To study how different initial T cell values, $T_0$, give different evolutions of D-dimer traces and thus influence the disease severity, we assume a linear dependence of T cell activation rate $\delta$ and inflammation inhibition rates, $\lambda$, $v_d$, $v_h$ with $T_0$. When $T_0 = 0.39$ (for non-survivors, determined from figure 4b), Eqn (13)–(16) give parameter values of non-survivors. When $T_0 = 0.84$ (for survivors, determined from figure 4b), Eqn (13)–(16) give survivors' parameter values. Simulation in figure 5c is

performed using equation (2.1)–(2.5) with a series of $T_0$ which gives the corresponding $\delta$, $\lambda$, $v_d$, $v_h$. Other parameters take fixed values as those of non-survivors.

$$\delta = 0.25 + 21.67 \times (T_0 - 0.39), \tag{5.4}$$
$$\lambda = 4.62 \times (T_0 - 0.39), \tag{5.5}$$
$$v_d = 0.889 \times (T_0 - 0.39) \tag{5.6}$$
and
$$v_h = 0.933 \times (T_0 - 0.39). \tag{5.7}$$

Data accessibility. Data and relevant code for this research work are stored in GitHub: https://github.com/luhaozhang/covid19_openscience and have been archived within the Zenodo repository: https://doi.org/10.5281/zenodo.5717666. The data are provided in electronic supplementary material [66].

Authors' contributions. L.Z.: data curation, investigation, software, validation, visualization, writing-original draft, writing-review & editing; R.L.: conceptualization, formal analysis, investigation, methodology, supervision, validation, visualization, writing-review & editing; G.S.: investigation, writing-review & editing; G.S.: funding acquisition, Writing-review & editing; Z.S.: conceptualization, methodology, project administration, supervision. All authors gave final approval for publication and agreed to be held accountable for the work performed therein.

Competing interests. The authors declare no competing interests.

Funding. The research is funded by the W. M. Keck Foundation through award no. 1005586.

Acknowledgements. The authors acknowledge Prof. Kelvin To for sharing data; Xiaoquan Wang, Guanghui She for discussion; Xinzi Zhang for writing suggestions.

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
