## [Peer Review File · Royal Society Open Science]

Review History

RSOS-211078.R0 (Original submission)

Review form: Reviewer 1

Is the manuscript scientifically sound in its present form?

Yes

Are the interpretations and conclusions justified by the results?

No

Is the language acceptable?

Yes

Do you have any ethical concerns with this paper?

No

Have you any concerns about statistical analyses in this paper?

No

Recommendation?

Reject

Comments to the Author(s)

I appreciate the work of the authors to clarify this interesting and potentially important work, but I still find the presentation quite difficult to follow, and the conclusions still too strong for the evidence as presented.

It remains unclear to me the appropriate time scale for this model. For example, making viral growth exponential, with only controls from the immune system, means that target cell limitation is irrelevant. Is there any evidence for this? Given how much the environment within the body changes during the entire course of infection, it is hard to imagine that the same dynamics are appropriate, particularly the emergence of T cells with no delay. The statement "the maximum variation of the T cell concentration is observed close to the viral load peak" is confusing. I don't see much of any peak in T cells in Figure 2 or in Figure 5.

The model makes many assumptions, which is of course essential, but the interpretation must be made carefully. As a couple of examples, on page 6, line 10, "common sense" is OK, but T cells expand from a pre-existing population of naive T cells that were there before the infection, and then expand through replication. Antibodies are modeled as killing viruses rather than blocking infection of new cells (which is missing from the model because target cells are not included). This could have a major effect because they are assumed to be recruited proportional to delayed virus numbers rather than infected cell numbers. I find it rather strange that IL-6 is produced proportional to antibodies rather than T cells, viruses or infected cells. As I said in my earlier review, every useful simple model has to make assumptions, but we have to be careful to interpret our conclusions in light of them.

I still find the presentation of the data very unclear. How are the 447 patients broken into four categories? Are mild patients the same as survivors?

Figure 3 is central to the paper, and makes the claim that T cells are much better than antibodies. Because these measurements are made after onset, is it possible that differences in T cell numbers are set before any measurements, and that T cells are the consequence of some early dynamics rather than being causal themselves? The statistic used is sensitive to the assumption that antibodies clear virus in the same way as T cells, even though T cells kill infected cells and antibodies inactivate (and mark) the virus particles themselves. I do not think that this paper does enough to warrant the conclusion on page 10, line 7, but instead presents an interesting and important hypothesis. Similarly, the conclusion on page 13 is consistent with the model, but hardly demonstrated given the number of correlated factors that underlie the T cell numbers. I like the use of this model to propose hypotheses about older patients and male patients, but describing how this would be distinguished from other hypotheses would provide more perspective on how to use the models.

Figure 4, in my opinion, should come much earlier, although I do not understand how Mild and Severe patients fit in. I'm troubled by the extrapolation of the IgG graph, and cannot understand the explanation. The observed data show no difference between survivors and non-survivors.

In the Discussion, the remarkable effectiveness of all the vaccines is not discussed. How much T cell immunity do they induce? Are they consistent with the models? I remain rather skeptical of GCSF, meditation (if that is what was meant on line 22) and other approaches for T cell stimulation. Finally, the list of assumptions is rather short. Trying to be more comprehensive does not make sense, but perhaps describing future experiments in detail would be more useful.

More minor points:

page 2, line 22: I think "monotonous" should be "monotonic".

page 2, line 32: This opening sentence is out of date and will be again when this is published. Everyone knows how important this pandemic is, and this is not needed.

page 3, line 52: I appreciate that the authors have worked on interesting methods related to the presentation here, but this model is a standard extension of a virus dynamics model and the reference to physics is not needed.

page 4, Figure 1: Is IL-1 the same as IL-1beta? And why is this mentioned in the legend given that it is not in the model (similarly for macrophages and NK cells)?

page 5, line 5: I don't see why the inflammation response would follow the immune response. Aren't they part of the same process? This whole paragraph has no citations to back up the choice of mechanisms.

page 5, line 42: How about "which incorporates both viral replication and viral clearance..."

page 7, line 34: How good a marker is CD3? The description of the T cell types after this was very hard for me to follow.

page 8, line 4: Isn't this just r^2 ? r^2 is not a good statistic for evaluating the quality of a model fit, because it does not take into account the number of parameters.

page 8, Figure 2: What is the justification for using T cell data from the severe group for patient 4?

page 8, line 51: Give the equation rather than stating it in words!

page 10, line 19: This is not a complete sentence, and the "demonstrates the validity" is far too strong a conclusion. The last sentence on this page adds uncertainty to the interpretation of IL-6, which I think needs to be better integrated with the other conclusions.

page 11, line 44: This correlation does not look at all significant, and I do not understand the source of the number of data points of each type.

page 11, line 48: Where are regulatory T cells in this model?

Figure 5: I mentioned some questions about panel a above, but why are there three symbols in Figure 5b but six items in the legend?

page 13, line 35: This should be phrased in a more balanced way.

page 15, line 32: I'm really confused by this classification as severe.

page 15, line 35: Would the source of data matter for the models? Are T cells and the other markers equally well represented?

page 16, line 10: Why are different scales used?

Review form: Reviewer 2

Is the manuscript scientifically sound in its present form?

Yes

Are the interpretations and conclusions justified by the results?

Yes

Is the language acceptable?

Yes

Do you have any ethical concerns with this paper?

No

Have you any concerns about statistical analyses in this paper?

No

Recommendation?

Accept as is

Comments to the Author(s)

See Appendix A.

Decision letter (RSOS-211078.R0)

Dear Miss She

The Editors assigned to your paper RSOS-211078 "Impairment of T cells' antiviral and anti-inflammation immunities dominates death from COVID-19" have made a decision based on their reading of the paper and any comments received from reviewers.

Regrettably, in view of the reports received, the manuscript has been rejected in its current form. However, a new manuscript may be submitted which takes into consideration these comments.

We invite you to respond to the comments supplied below and prepare a resubmission of your manuscript. Below the referees' and Editors' comments (where applicable) we provide additional requirements. We provide guidance below to help you prepare your revision.

Please note that resubmitting your manuscript does not guarantee eventual acceptance, and we do not generally allow multiple rounds of revision and resubmission, so we urge you to make every effort to fully address all of the comments at this stage. If deemed necessary by the Editors, your manuscript will be sent back to one or more of the original reviewers for assessment. If the original reviewers are not available, we may invite new reviewers.

Please resubmit your revised manuscript and required files (see below) no later than 13-Feb-2022. Note: the ScholarOne system will 'lock' if resubmission is attempted on or after this deadline. If

you do not think you will be able to meet this deadline, please contact the editorial office immediately.

Please note article processing charges apply to papers accepted for publication in Royal Society Open Science (<https://royalsocietypublishing.org/rsos/charges>). Charges will also apply to papers transferred to the journal from other Royal Society Publishing journals, as well as papers submitted as part of our collaboration with the Royal Society of Chemistry (<https://royalsocietypublishing.org/rsos/chemistry>). Fee waivers are available but must be requested when you submit your manuscript (<https://royalsocietypublishing.org/rsos/waivers>).

Thank you for submitting your manuscript to Royal Society Open Science and we look forward to receiving your resubmission. If you have any questions at all, please do not hesitate to get in touch.

on behalf of Professor Tim Rogers (Associate Editor) and Mark Chaplain (Subject Editor)
openscience@royalsociety.org

Associate Editor Comments to Author (Professor Tim Rogers):

I agree with the main point of Referee 1 - the evidence presented is nowhere near strong enough to justify the conclusions made. If you choose to resubmit, I suggest you rewrite the manuscript (including the title!) to make clear the limitations of the study, and phrase your conclusions as a call to investigate your hypothesis in more detail rather than as conclusive proof.

Reviewer comments to Author:

Reviewer: 1

Comments to the Author(s)

I appreciate the work of the authors to clarify this interesting and potentially important work, but I still find the presentation quite difficult to follow, and the conclusions still too strong for the evidence as presented.

It remains unclear to me the appropriate time scale for this model. For example, making viral growth exponential, with only controls from the immune system, means that target cell limitation is irrelevant. Is there any evidence for this? Given how much the environment within the body changes during the entire course of infection, it is hard to imagine that the same dynamics are appropriate, particularly the emergence of T cells with no delay. The statement "the maximum variation of the T cell concentration is observed close to the viral load peak" is confusing. I don't see much of any peak in T cells in Figure 2 or in Figure 5.

The model makes many assumptions, which is of course essential, but the interpretation must be made carefully. As a couple of examples, on page 6, line 10, "common sense" is OK, but T cells expand from a pre-existing population of naive T cells that were there before the infection, and then expand through replication. Antibodies are modeled as killing viruses rather than blocking infection of new cells (which is missing from the model because target cells are not included).

This could have a major effect because that are assumed to be recruited proportional to delayed virus numbers rather than infected cell numbers. I find it rather strange that IL-6 is produced

proportional to antibodies rather than T cells, viruses or infected cells. As I said in my earlier review, every useful simple model has to make assumptions, but we have to be careful to interpret our conclusions in light of them.

I still find the presentation of the data very unclear. How are the 447 patients broken into four categories? Are mild patients the same as survivors?

Figure 3 is central to the paper, and makes the claim that T cells are much better than antibodies.

Because these measurements are made after onset, is it possible that differences in T cell numbers are set before any measurements, and that T cells are the consequence of some early dynamics rather than being causal themselves? The statistic used is sensitive to the assumption that antibodies clear virus in the same way as T cells, even though T cells kill infected cells and antibodies inactivate (and mark) the virus particles themselves. I do not think that this paper does enough to warrant the conclusion on page 10, line 7, but instead presents an interesting and important hypothesis. Similarly, the conclusion on page 13 is consistent with the model, but hardly demonstrated given the number of correlated factors that underlie the T cell numbers. I like the use of this model to propose hypotheses about older patients and male patients, but describing how this would be distinguished from other hypotheses would provide more perspective on how to use the models.

Figure 4, in my opinion, should come much earlier, although I do not understand how Mild and Severe patients fit in. I'm troubled by the extrapolation of the IgG graph, and cannot understand the explanation. The observed data show no difference between survivors and non-survivors.

In the Discussion, the remarkable effectiveness of all the vaccines is not discussed. How much T cell immunity do they induce? Are they consistent with the models? I remain rather skeptical of GCSF, meditation (if that is what was meant on line 22) and other approaches for T cell stimulation. Finally, the list of assumptions is rather short. Trying to be more comprehensive does not make sense, but perhaps describing future experiments in detail would be more useful.

More minor points:

page 2, line 22: I think "monotonous" should be "monotonic".

page 2, line 32: This opening sentence is out of date and will be again when this is published. Everyone knows how important this pandemic is, and this is not needed.

page 3, line 52: I appreciate that the authors have worked on interesting methods related to the presentation here, but this model is a standard extension of a virus dynamics model and the reference to physics is not needed.

page 4, Figure 1: Is IL-1 the same as IL-1beta? And why is this mentioned in the legend given that it is not in the model (similarly for macrophages and NK cells)?

page 5, line 5: I don't see why the inflammation response would follow the immune response. Aren't they part of the same process? This whole paragraph has no citations to back up the choice of mechanisms.

page 5, line 42: How about "which incorporates both viral replication and viral clearance..."

page 7, line 34: How good a marker is CD3? The description of the T cell types after this was very hard for me to follow.

page 8, line 4: Isn't this just r^2 ? r^2 is not a good statistic for evaluating the quality of a model fit, because it does not take into account the number of parameters.

page 8, Figure 2: What is the justification for using T cell data from the severe group for patient 4?

page 8, line 51: Give the equation rather than stating it in words!

page 10, line 19: This is not a complete sentence, and the "demonstrates the validity" is far too strong a conclusion. The last sentence on this page adds uncertainty to the interpretation of IL-6, which I think needs to be better integrated with the other conclusions.

page 11, line 44: This correlation does not look at all significant, and I do not understand the source of the number of data points of each type.

page 11, line 48: Where are regulatory T cells in this model?

Figure 5: I mentioned some questions about panel a above, but why are there three symbols in Figure 5b but six items in the legend?

page 13, line 35: This should be phrased in a more balanced way.

page 15, line 32: I'm really confused by this classification as severe.

page 15, line 35: Would the source of data matter for the models? Are T cells and the other markers equally well represented?

page 16, line 10: Why are different scales used?

Reviewer: 2

Comments to the Author(s)

N/A.

===PREPARING YOUR MANUSCRIPT===

While not essential, it will speed up the preparation of your manuscript proof if accepted if you format your references/bibliography in Vancouver style (please see

<https://royalsociety.org/journals/authors/author-guidelines/#formatting>). You should include DOIs for as many of the references as possible.

===PREPARING YOUR REVISION IN SCHOLARONE===

Author's Response to Decision Letter for (RSOS-211078.R0)

See Appendix B.

RSOS-211606.R0

Review form: Reviewer 2

Is the manuscript scientifically sound in its present form?

Yes

Are the interpretations and conclusions justified by the results?

Yes

Is the language acceptable?

Yes

Do you have any ethical concerns with this paper?

No

Have you any concerns about statistical analyses in this paper?

No

Recommendation?

Accept as is

Comments to the Author(s)

The authors have put in a considerable amount of effort to review the paper and respond to reviewer comments which is appreciated.

Decision letter (RSOS-211606.R0)

Dear Miss She

On behalf of the Editors, we are pleased to inform you that your Manuscript RSOS-211606 "Impairment of T cells' antiviral and anti-inflammation immunities is critical to death from COVID-19" has been accepted for publication in Royal Society Open Science subject to minor revision in accordance with the referees' reports. Please find the referees' comments along with any feedback from the Editors below my signature.

Please submit your revised manuscript and required files (see below) no later than 7 days from today's (ie 17-Nov-2021) date. Note: the ScholarOne system will 'lock' if submission of the revision is attempted 7 or more days after the deadline. If you do not think you will be able to meet this deadline please contact the editorial office immediately.

on behalf of Professor Tim Rogers (Associate Editor) and Mark Chaplain (Subject Editor)
openscience@royalsociety.org

Associate Editor Comments to Author (Professor Tim Rogers):
Associate Editor
Comments to the Author:

I believe that the authors have addressed the referees comments well everywhere except in the title. Please could they consider changing "is critical" to "may be critical".

Reviewer comments to Author:
Reviewer: 2
Comments to the Author(s)

The authors have put in a considerable amount of effort to review the paper and respond to reviewer comments which is appreciated.

===PREPARING YOUR MANUSCRIPT===

one version should clearly identify all the changes that have been made (for instance, in coloured highlight, in bold text, or tracked changes);

===PREPARING YOUR REVISION IN SCHOLARONE===

-- If you are requesting an article processing charge waiver, you must select the relevant waiver option (if requesting a discretionary waiver, the form should have been uploaded, see 'File upload' above).

-- If you have uploaded any electronic supplementary (ESM) files, please ensure you follow the guidance at <https://royalsociety.org/journals/authors/author-guidelines/#supplementary-material> to include a suitable title and informative caption. An example of appropriate titling and captioning may be found at https://figshare.com/articles/Table_S2_from_Is_there_a_trade-off_between_peak_performance_and_performance_breadth_across_temperatures_for_aerobic_scope_in_teleost_fishes_/3843624.

Author's Response to Decision Letter for (RSOS-211606.R0)

See Appendix C.

Decision letter (RSOS-211606.R1)

Dear Miss She,

I am pleased to inform you that your manuscript entitled "Impairment of T cells' antiviral and anti-inflammation immunities may be critical to death from COVID-19" is now accepted for publication in Royal Society Open Science.

COVID-19 rapid publication process:

We are taking steps to expedite the publication of research relevant to the pandemic. If you wish, you can opt to have your paper published as soon as it is ready, rather than waiting for it to be published the scheduled Wednesday.

This means your paper will not be included in the weekly media round-up which the Society sends to journalists ahead of publication. However, it will still appear in the COVID-19 Publishing Collection which journalists will be directed to each week (<https://royalsocietypublishing.org/topic/special-collections/novel-coronavirus-outbreak>).

If you wish to have your paper considered for immediate publication, or to discuss further, please notify openscience_proofs@royalsociety.org and press@royalsociety.org when you respond to this email.

on behalf of Professor Tim Rogers (Associate Editor) and Mark Chaplain (Subject Editor)
openscience@royalsociety.org

Appendix A

Overall comments:

In this paper, the authors develop a mathematical model for the immune response to SARS-Cov-2. They match their model with human data and investigate the impact of T cells on the survival of COVID-19 patients. They conclude that the initial number of T cells could be a major cause of COVID death and that the main trigger of a cytokine storm is the secretion of elevated non-neutralizing antibodies. The paper's strengths are that the code/data is available, and that the model developed is simple (only 19 parameters) and is fit to clinical data. The limitations, however, are that the paper is unclear in the model assumptions and lack's supporting evidence in a lot of places for the findings. In it's current state, I also don't think the paper would be reproducible and there are areas where aspects of the work need to be clarified.

I've listed my comments into major and minor. If the authors are able to make changes and respond to my comments I'd be happy to consider a second review of the paper. In addition, the language in the paper is poor with numerous grammatically incorrect statements. A through edit of the paper's language is necessary.

Major comments:

- The authors propose that the “quantitative role of (T cells and antibodies) in antiviral and anti-inflammatory immune responses is unknown” and then further state that a key questions is “what is the relative importance of T cell and antibody response for antiviral immunity at the different stages”. However, the role of T cells and antibodies in terms of clearing a viral infection is well known in immunology. In turn, it is well known in COVID-19 patients that T cell numbers are decreased with more severe disease. Can the authors be more specific on that they are suggesting to evaluate? Or better motivate why they believe that the role of T cells and antibodies in antiviral and anti-inflammatory immune responses in COVID is unknown.
- Can the authors explain what they mean by “top-down modelling”? The authors mention this is related to “order functions” but not proper explanation is given as to what they mean by this approach
- The variables in the model V, T_e, A, I, S_i are not defined when the model is described, they are first described in the results and then methods section. A description of each of the variables should be given when the assumptions of the model are described (with the equations) so the reader can understand the model. In addition, explanation for the terms in the model should be given with supporting evidence from the literature. A schematic with the model variables may also help the reader (can be included in the SI). I have listed below a few of my concerns about the assumptions:
 - Can the authors justify why there would be continuous viral replication in the body? In reality, virus replication depends on infected cells, so can the authors support this assumption either mathematically (i.e. with other models in the literature) or biologically. I see the authors say this is to account for “antiviral effect of innate immunity” but it's not clear what this is or how this terms accommodates for that.
 - Can the authors explain why the neutralizing effect of antibodies on virus is proportion to the difference between the number of antibodies and the initial antibody concentration? This is the term $\gamma*(A-A_0)*V$. This is if I am reading the model correctly.

- Why is there a delay in antibody production based on virus concentration? This needs to be explained and justified
- Is IL-6 production proportional to antibody concentration? i.e. the term $\kappa \cdot A$. I don't think this could be the case since IL-6 is known to be produced much earlier during inflammation and is the primary response, whereas antibody concentrations are well known to peak later in inflammation. This needs to be justified and supported by evidence in the literature.
- The method through which the upper and lower bounds for the parameters in Table S4 were obtained is not clear to me, can the authors explain this further when they discuss the uncertainty of parameters in their model and how this relates to parameter identifiability.
- In the methods, please describe the data in more details e.g. viral load over 20 days measure in sera, IL-6 concentrations over time in plasma etc. The current description is not detailed enough or clear enough, for example, what do the authors mean by "The CD3+T cell data of mild, severe and critical population,..., where used for individual patients and groups of the same severity who don't have T cell data". What is meant by "individual patients and groups"? Also is there meant to be a column for the IL-6 and D-dimer data?
- Can the authors explain why, in Figure 2, the T cells numbers decrease in the mild and severe cases from the initial concentration. Why do the T cells numbers go down and can this be shown to be true in the literature? My intuition would have said that in a mild infection there would be more activated T cells than before infection started (due to T cell proliferation from antigen stimulation). In addition, can the other variables in the model be plotted for the dynamics in Figure 2 (even in a supplementary figure) so we can see what role the macrophages/NKs and D-dimer are playing in this model?
- Can the authors comment on why the IL-6 concentration in non-survivors is significantly higher initially than the IL-6 concentration in survivors? I would have thought that initially the concentration in non-survivors and survivors would have been the same. Also the fit of the survivors IL-6 concentration looks different to the data, concave up vs concave down. Can the authors explain what they think might be happening here?
- A more rigorous exploration of treatment effects in the model is needed to say that this work "points out a new direction to advance current prevention and treatment" as I don't feel this was demonstrated in the model results. Can the authors simulate treatments and show how the model predicts different efficacies or at least further elaborate on how their results support this point.

Minor comments:

- grammar of the article needs to be improved, I've given three examples below but a significant read through and edit is necessary:
 - "Here, we adopt top-down modelling..." -> "Here, we adopt a top-down modelling..."
 - "Past clinical study has found a series of host factors..." -> "Past clinical studies have found a series of host factors..."
 - "resulting in several unsolved questions about the cause and saving of the death" -> "saving of the death" is not the correct expression
- A lot of the mathematical modelling referenced by the author on COVID-19 immune related dynamics are classified as an over-simplified or over-complicated version of the dynamics. Ther

authors should be more specific about what is particularly lacking in these “over-simplified” models that has then been included in their model, as their model is quite simple. I also think, that since this model is claimed to be one of the first that considers “organ damages and disease progression” it might be worth referencing this paper. While much more complicated than the author’s model, it does incorporate these aspects that the authors are proposing to model:

<https://doi.org/10.1101/2021.01.05.425420>

- Typo: should be a space in “interactionNAb”
- There are no references for the biology described in lines 34-53 of page 5
- The parameters n_T needs to be defined after equation (10), as well as n_{sd} , n_I , n_{Sh} and so on..
- I think the caption of Figure S1 has labelled T cells and antibodies incorrectly
- In caption of Figure 2, what is P1, P2 and P3? Also how were the normal ranges (black dotted lines) determine? Reference? Also why is Anti-S1 IgG?
- Change the axis in Fig 3c so we can see the fit to the data better
- Please provide the statistics for Figure 4a
- The colour scheme in Figure 4c does not qualitatively provide a difference between the D-dimer as a function of T cell numbers. Please pick a different colour scheme. Try colorbrewer.

Appendix B

Response to referee:

We appreciate your thoughtful comments, which lead to substantial improvement of the main text, including more careful presentations of our assumptions and more precise conclusions and clearer interpretations and discussions in detail. Below, we respond, point-by-point, to all comments (the black is the referee's comments, and the red is our reply).

Associate Editor Comments to Author (Professor Tim Rogers):

I agree with the main point of Referee 1 - the evidence presented is nowhere near strong enough to justify the conclusions made. If you choose to resubmit, I suggest you rewrite the manuscript (including the title!) to make clear the limitations of the study, and phrase your conclusions as a call to investigate your hypothesis in more detail rather than as conclusive proof.

Reply: Following your suggestion, we have rewritten the manuscript.

- (1) We have revised our title to be “Impairment of T cells’ antiviral and anti-inflammation immunities is critical to death from COVID-19”, which emphasize T cells’ importance rather than their leading role.
- (2) We have rephrased our conclusions more precisely as “supports our hypothesis” and “important inferences based on our assumptions of the immunodynamics (i.e., Eq.(1-5))” rather than proved conclusions. For instance, we have revised the finding on the last paragraph on page 10 and the last paragraph on page 11 to be, “The consistency between simulation and data supports our hypothesis about the antiviral dynamics in which adaptive response plays a significant role...” and “This finding reveals an important inference in COVID-19 that T cells may have a dominant role in the virus clearance relative to antibodies, especially for mild patients.” Also, we have revised the finding on page 12th to be “The agreement between the experiment and the data supports the validity of Eq. (1)-(5)”. Besides, we have revised the main conclusion on page 15 to be “First of all, this model provides an accurate description of real-time clinical data involving hundreds of patients, which then support T cells' critical roles in the antiviral and anti-inflammatory immune responses of COVID-19. Furthermore, beyond the previous correlation analysis for T cell scarcity and disease severity, a causal relationship between death from COVID-19 and impaired T cell immunity is supported by this work, providing a preliminary explanation for the high mortality of older men.”
- (3) We have proposed some further researches to verify these clinically important inferences as follows: “It is worth mentioning that these important findings are

inferences based on our assumptions of the immunodynamics (i.e., Eq. (1-5)) and are preliminarily supported by the present work. Therefore, considering the strong clinical needs, there is an urgent call to investigate them in more detail. For instance, we suggest two further studies to verify whether the T cells play a dominant role in adaptive immunity. On the one hand, Eq. (1)-(3) should be applied to more clinical data of individual patients, which should be measured for viral load, effector T cells and neutralizing antibodies simultaneously. On the other hand, we suggest carrying out some animal experiments with COVID-19 to compare the effectiveness of vaccines or drugs that activate (or suppress) antibody and T cells immunity, respectively.”

- (4) We have made a clear statement of the entire course of immunodynamics of COVID-19, the stage that this work focuses on, the reason neglecting target cell limitation and T cell delay, and the time scale of the model.
- (5) We have entirely rewritten the model description with a clearer description of the physiological pathways for every term and the corresponding assumptions that lead to the mathematical forms.
- (6) We have explained more clearly how the data of effector T cells come from the reduction of CD3+ T cells in the blood and revised the y axis of Figure 2 to be effector T cells to present more clearly the “T cell peak”.
- (7) We have explained more clearly how data from different sources are integrated to approximate the data of individual patients and mild, severe, survivor and non-survivor groups.
- (8) In the Discussion, we have added vaccine effectiveness and its consistency with the model, as well as the references for T cell stimulation by meditation and health activities.
- (9) We have made a clearer statement of the distinction between our physical reference and the traditional virus dynamics model.

Reviewer: 1

Comments to the Author(s)

I appreciate the work of the authors to clarify this interesting and potentially important work, but I still find the presentation quite difficult to follow, and the conclusions still too strong for the evidence as presented.

Reply: Thanks for your appreciation. And we agree that the presentation can be further revised to be easier to follow, and the conclusions should be rephrased to exhibit our findings more accurately.

It remains unclear to me the appropriate time scale for this model. For example, making viral growth exponential, with only controls from the immune system, means that target cell limitation is irrelevant. Is there any evidence for this? Given how much the environment within the body changes during the entire course of infection, it is hard to imagine that the same dynamics are appropriate, particularly the emergence of T cells with no delay. The statement "the maximum variation of the T cell concentration is observed close to the viral load peak" is confusing. I don't see much of any peak in T cells in Figure 2 or in Figure 5.

Reply: Thanks for your in-depth comments about the basic assumptions of the model and the main aim of this paper.

First of all, we choose to model a dynamic with the basic time scale of days. In this modelling, the doubling time of the viral load equals to $\frac{\ln 2}{\alpha} = 1.0 \pm 0.6$ days, meaning the virus takes 1.0 ± 0.6 days to double the amount by replication, which is consistent with the slope of viral load (log scale) in the increasing stage (Fig.4 in Ref¹). The T cells have a half-life equal to $\frac{\ln 2}{\epsilon} = 11.2 \pm 9.8$ days, which is also consistent with the general observation that most effector T cells undergo apoptosis following viral clearance². The antibodies have a half-life equal to $\frac{\ln 2}{\theta} = 35 \pm 22$ days (decline by 50%), consistent with the recent observation of longitudinal decline of neutralizing antibodies by more than 90% after 60-100 days post onset³.

These results are summarized as a new paragraph on page 10.

Secondly, we agree that whether target cell limitation is engaged in controlling viral dynamics should be addressed. At present, although the target cell limited model is applied to COVID-19¹, there is no direct experimental evidence showing the target cells are depleted when the viral load of SARS-CoV-2 peaks. Therefore, we choose to ignore the effect of target cell limitation and claim that the T-cell contribution is essential for the viral load peak, which is new but needs further verification in the future.

Thirdly, we agree that the immunodynamics of COVID-19 has multiple stages. Specifically, we think it can be divided into three stages, and this work focuses on the second stage. The first stage is the early stage of infection when the adaptive response has not been activated, and only the physical barrier (skin, mucosa, cilium) and the innate response is actively protecting the body. At the second stage, adaptive response produces virus-specific T cells and antibodies interacting against the virus. The third stage is the dying

stage of non-survivors; the virus expands to the whole body, the target cells may be depleted, and protective immune pathways are massively blocked.

We believe that the success or failure of the virus-host battle during the second stage determines whether the patients can survive or not, so we focus on the second stage in the present work. We assume that, in this stage, the viral load is growing exponentially due to sufficient target cells, and virus-specific T cells and antibodies are produced significantly to suppress the viral load. Therefore, we neglect the slight initial delay of the effector T cell production (i.e., the activation of naïve T cells) and the ultimate target cell limitation. Quantitatively, the immunodynamics in the second stage are modelled in bilinear kinetics, as commonly done¹ and the epidemiology⁴. It turns out, surprisingly, in our study that such modelling is consistent with most clinical data. Furthermore, in our three-stage perspective, we suspect that the current model may be modified for asymptomatic patients and the dying stage of non-survivors by including significant innate response and effect from target cell limitation. For instance, the temporal evolution of the innate response may explain the slight deviation of the simulation from IL-6 dynamics data of survivors on the 7th~13th day (shown in Fig.4(d)).

Finally, "there is no peak of effective T cells" is a misunderstanding, which owes to our bad explanation and presentation of T cell data in Figure2 and Figure4. Therefore, we have revised the text of T cell data explanation in the first paragraph as: "Because it's difficult to detect the effective T cell number inside organs, we estimate the effective T cell number from the experimentally measured CD3+ T cell data in serum. Zhang et al.⁵ deduced that the reduction of CD3+ in serum represents the number of cells that entered organs and became effector T cells. In this context, we define, at the present stage, the amount of effector T cell is proportional to the reduction of CD3+ in serum." On the other hand, in order to present the data more clearly, we have revised the y-axis quantity in figure2 and figure4 from CD3+ number in serum to the estimated effective T cell number. After this revision, one can clearly see that the effective T cells have a significant peak (for example, around the 8th day in Fig.2a) and is closed to the viral peak (around the 4th day in Fig.2a). Hence, the T-cell maximum close to the viral-load peak is not confusing but an observed fact.

Based on the above discussions, we have added a second paragraph on page 5 to discuss specifically the scope that the model can be applied as well as a paragraph on page 17 ("Discussion") for the improvement of the model for asymptomatic patients and dying patients.

The model makes many assumptions, which is of course essential, but the interpretation must be made carefully. As a couple of examples, on page 6, line 10, "common sense" is OK, but T cells expand from a pre-existing population of naive T cells that were there before the infection, and then expand through replication. Antibodies are modeled as killing viruses

rather than blocking infection of new cells (which is missing from the model because target cells are not included). This could have a major effect because that are assumed to be recruited proportional to delayed virus numbers rather than infected cell numbers. I find it rather strange that IL-6 is produced proportional to antibodies rather than T cells, viruses or infected cells. As I said in my earlier review, every useful simple model has to make assumptions, but we have to be careful to interpret our conclusions in light of them.

Reply: We agree that we have not carefully explained the assumptions of our model, which may cause puzzling. Therefore, with the reviewer's points addressed, we have entirely rewritten the model description with a clearer statement of the physiological pathways for every term, see the new section "Causal network and mathematic model of the Antiviral-Inflammation responses". The critical point is that our model is a concise macroscopic model, which focuses on the dynamics of six main elements in the virus-immune system battle and synthesizes a large number of microscopic physiological processes into the activation and inhibition coefficients of these dynamics. Therefore, we have elaborated more on the corresponding activation and inhibition coefficients, which we missed before, in the main text. We summarize the answers to the reviewer's points in the next paragraph.

We agree it's clearer to discuss initial T cell concentration with the background knowledge of pre-existing naïve T cells' expansion, which is added in the fourth paragraph (effector T cell dynamics) in the "Causal network and mathematic model of the Antiviral-Inflammation responses" section, as: "initially naïve CD8+ T cells are first activated by the free virus through antigen presentation by macrophages, dendritic cells and CD4+ cells, or directly by infected cells⁷. Naïve CD8+ cells then differentiate, go through clonal expansion into effector T cells¹⁰." Besides, we added justification for the approximation that initial T cell concentration is 0 in the last paragraph in this section as: "according to Eq. (2), the production rate of effector T cells, is the maximum when the virus peaks and is several magnitudes smaller on t_0 (the beginning day of the simulation) when the virus is low. Therefore, we approximate the initial effector T cell and antibody concentration to zero because there is no virus-specific effective T cell or antibody before infection⁶."

We agree that antibody blocks infection of new cells, which has been added in the main text in the first paragraph on page 6. However, it is worth mentioning that the model is aimed at quantifying the total amount reduction of the virus by antibodies using simple mathematics (i.e., the bilinear dynamics) rather than a clarification of the physiological details. Specifically, the bilinear term $-\gamma A(t)V(t)$ integrates the whole process that the neutralizing antibodies bind to the free virus, preventing the virus from entering the cells, and the neutralized virus is cleared by other macrophages, so the virus decreases. Furthermore, the production of neutralizing antibodies should not be assumed proportional to infected cell numbers but virus number because it is the virus, not infected cells, that serve as the source to promote antibody production: naïve B cells are activated directly by

free virus or indirectly through antigen presentation by macrophages, dendritic cells and CD4+ cells⁷.

It is suggested⁸ that non-neutralizing antibodies bind to macrophages and promote macrophages susceptible to virus, and accompanied with this process, macrophages secrete a large amount of IL-6. Based on this, we propose that IL-6 production by non-neutralizing antibodies dominates the cytokine storm taking place at later stage⁹, and neglect the IL-6 produced by T cells, monocytes and infected cells¹⁰. Therefore, we assumed that the production rate of IL-6 is proportional to the concentration of non-neutralizing antibodies, which is assumed proportional to the concentration of neutralizing antibodies.

I still find the presentation of the data very unclear. How are the 447 patients broken into four categories? Are mild patients the same as survivors?

Reply: We are sorry about the unclear presentation. So, we have revised the main text for a more transparent presentation. Specifically, to describe the coupled evolutions of the virus, effector T cells, antibodies, IL-6, D-dimer and HSCT, we have to use a complete set of data for all these six quantities; however, there is no literature that reports such a complete set in one time. Therefore, we use data from multiple sources to construct the complete data set. The data set are constructed from data sources of the same severity, and there are in total four data sets: mild, severe, survivor and non-survivor group.

Figure 3 is central to the paper, and makes the claim that T cells are much better than antibodies. Because these measurements are made after onset, is it possible that differences in T cell numbers are set before any measurements, and that T cells are the consequence of some early dynamics rather than being causal themselves?

The statistic used is sensitive to the assumption that antibodies clear virus in the same way as T cells, even though T cells kill infected cells and antibodies inactivate (and mark) the virus particles themselves.

I do not think that this paper does enough to warrant the conclusion on page 10, line 7, but instead presents an interesting and important hypothesis. Similarly, the conclusion on page 13 is consistent with the model, but hardly demonstrated given the number of correlated factors that underlie the T cell numbers.

I like the use of this model to propose hypotheses about older patients and male patients, but describing how this would be distinguished from other hypotheses would provide more perspective on how to use the models.

Reply: Thanks for your in-depth comments. First, we agree there are early dynamics for T cell activation before any measurements. Then, using our model, we can predict early T cells by fitting the T cell data after onset, as shown in Figure 2j and Figure S2a (3rd line). We suggest further measurements of T cells in latent patients to verify these predictions. We have added this discussion as a third paragraph on page 10.

Secondly, we agree that the essence of our hypothesis is that both the process of antibody inactivating free virus and the process of T cells killing infected cells follow bilinear kinetics; that is, the virus clearance rate by T cells and antibodies are proportional to the product of total viral concentration (proportional to the number of infected cells or free virus, in a statistical sense) and the concentration of antibodies or T cells. This bilinear kinetics is typical for the mathematical modelling of both the immune system¹ and the epidemiology⁴. We believe that this assumption is reasonable for the second stage of the virus-host battle when there is a significant adaptive response (i.e., virus-specific T cells and antibodies) fighting against the virus.

Therefore, we have made a statement before the comparison as follows: "Eq. 1 has simplified the antiviral responses of adaptive immune processes, i.e., T cells kill infected cells and antibodies inactivate the virus particles themselves to common bilinear kinetics. One advantage of this simplification is that it enables us to quantify (although preliminarily) the roles of T cells and antibodies in the antiviral process."

Thirdly, we agree with your rigorous comments about our statistics and conclusions; that is, we should regard these conclusions in the original draft as important inferences based on our assumptions of immunodynamics (i.e., Eq. (1-5)) and supported by data, rather than proved conclusions. Therefore, we rephrase these conclusions to be more precise. For instance, we have revised the original page 10, line 7 to be, "This finding reveals an important inference in COVID-19 that T cells may have a dominant role in the virus clearance relative to antibodies, especially for mild patients." which is now the end of the first paragraph on page 12.

Besides, we have revised the first paragraph (conclusion) in the "Discussion" section to be "First of all, this model provides an accurate description of real-time clinical data involving hundreds of patients, which then support T cells' critical roles in the antiviral and anti-inflammatory immune responses of COVID-19. Furthermore, beyond the previous correlation analysis for T cell scarcity and disease severity^{8,9}, this work reveals a causal relationship between death from COVID-19 and impaired T cell immunity provides a preliminary explanation for the high mortality of older men."

Besides, following your suggestion, we have proposed some further research plans to verify these clinically important inferences as follows (in the second paragraph of the "Discussion"): "It is worth mentioning that these important findings are inferences based on

our assumptions of the immunodynamics (i.e., Eq. (1-5)) and are preliminarily supported by the present work. Therefore, considering the strong clinical needs, there is an urgent call to investigate them in more detail. In other words, Eq. (1)-(5), the associated assumptions and the critical role of T cells need to be tested by more real-time data. For instance, we suggest further studies on two aspects. One is to verify whether the T cells play a dominant role in adaptive immunity. On the one hand, Eq. (1)-(3) should be applied to more clinical data of individual patients, for which viral load, CD8+ and neutralizing antibodies should be measured simultaneously. On the other hand, it is important to compare the effectiveness of drugs or vaccines that activate (or suppress) antibody and T cell immunity, respectively. The other is to verify whether the production of IL-6 is dominated by the combination of non-neutralizing antibodies and macrophages. In this context, we propose to measure real-time data of non-neutralizing antibodies, macrophages and IL-6 in clinical patients and then quantify their real-time correlations. We anticipate, at the early stage of the cytokine storm, the secretion rate of IL-6 would be proportional to the product of macrophage and non-neutralizing antibodies concentrations, i.e., the first term on the right-hand side of Eq. (4). This kind of verification can be applied to other assumptions as well, such as whether viral clearance rate from the innate response is constant around symptom onset and whether CD8+ concentration in the organ is proportional to the reduction of CD3+ concentration in peripheral blood."

Finally, following your suggestion, we have revised our hypotheses about older patients and male patients as follows (the first paragraph on page 15): "Other hypotheses ¹¹, for example, states that men produce lower levels of antibodies than women. According to our model, T cells are expected to have higher impacts on viral clearance than antibodies; therefore, lower levels of antibodies are less likely to cause worse protection against the virus than lower levels of T cells for men. We propose to measure dynamics of effector T cells, antibodies, and viral load in women and men and calculate the fractions of virus cleared by T cells (F_T) and antibodies (F_A) using the model. If antibodies are less protective in men as the other hypothesis states, we expected to see a lower F_A in men than women. If it is the T cells that are less protective in men as suggested by our model, we expect to see a lower F_T and higher F_A in men than women."

Figure 4, in my opinion, should come much earlier, although I do not understand how Mild and Severe patients fit in. I'm troubled by the extrapolation of the IgG graph, and cannot understand the explanation. The observed data show no difference between survivors and non-survivors.

Reply: Our understanding is that you think that we should present a complete simulation of all the six variables in the beginning and then discuss each part. We agree this would make the presentation more direct; however, due to the lack of a complete data set for any individual patient, we cannot do this for individual patients at present.

Therefore, in figure 2, we use the data of individuals (the P1, P3 and P4 patients) from a single source to present a rigorous verification of the antiviral model, which we think should come first. In contrast, the data in figure 4 is the data of groups, and they are from multiple sources that complement each other to construct a complete data set for all six variables. Therefore, we think it should come later. Besides, there is no mild and severe group in Figure 4, for we do not have these data; as data become more abundant, more verification can be done in the future.

We agree that we cannot estimate the saturation value of the observed IgG data of survivors and non-survivors solely because the data is incomplete. However, based on a complete measurement of critical and non-critical patients², we use the ratio of the saturation values of critical and non-critical patients to approximate the ratio of the saturation values of non-survivors and survivors, which guides the simulation of IgG in Figure 4. Therefore, we have added this discussion to the main text and revised the legend of Figure 4.

In the Discussion, the remarkable effectiveness of all the vaccines is not discussed. How much T cell immunity do they induce? Are they consistent with the models? I remain rather skeptical of GCSF, meditation (if that is what was meant on line 22) and other approaches for T cell stimulation. Finally, the list of assumptions is rather short. Trying to be more comprehensive does not make sense, but perhaps describing future experiments in detail would be more useful.

Reply: Thanks for providing these comments and suggestions, which inspire us to deepen our discussion to connect with literature and future experiments more closely.

First, currently, there are several widely used vaccines, e.g., BNT162b2 (Pfizer–BioNTech, two doses), AZD1222 (Astra Zeneca–University of Oxford, two doses), Ad26.COVID-2-S (Johnson & Johnson, one dose). The reported impact of these vaccine efficacies on SARS-CoV-2 variants are: BNT162b2 showed 90% efficacy for Alpha and 75% for Beta; AZD1222 showed 75% on Alpha and 10% for Beta; Ad26.COVID-2.S shows 70% for Alpha, and for Beta, it showed 72% efficacy in the USA, 66% in Latin America and 57% in South Africa. Especially, Ad26.COVID-2-S showed 85% effectiveness in preventing severe cases across the USA, Latin America and South Africa with one dose vaccine and sustained (and increasing) immune protection over time, which is suspected by former FDA Commissioner Scott Gottlieb, MD. to come from a robust early induction of memory T-cells (CD4+ and CD8+)¹². This is consistent with the model and is supported by the recent observation that T memory cells sustained in convalescent patients for more than ten months¹³ and virus-specific T cells are found even in uninfected people^{6, 14}. Relative to the short-term decay (decline by 90% after 60~100 days post onset³) of antibodies, the long-term existence of T-cell memory cells makes the development of vaccines associated with T cells activation have the

advantage of potential long-term protection, so it has become one of the main goals of future vaccine development^{13–16}.

Second, although we know that these methods are not in the mainstream yet, there is growing evidence that they may stimulate the immunity of T-cell. For instance, there is some recent evidence showing Chinese herbal medicine can improve the number and function of different T cell subsets, like CD4+ and CD8+, see recent review by Robert D. Hoffman and reference therein. Specifically, for COVID-19 patients, it's been found herbal medicine can obviously improve lymphocytes and shows remarkable therapeutic effects, like Shufen JieDu^{17,18}, and others reported from clinical treatment¹⁹. Furthermore, the mouse model found Shufen JieDu improves CD4+ and CD8+, significantly reduces the virus load in the lung from 1109.29 ± 696.75 to 0 ± 0 copies/mL, and reduce the cytokine level²⁰. Similarly, TaiChiQuan and Meditation have been found to have the effect of increasing CD4+ T cells from several reports, respectively^{21,22}. However, we have removed the discussion about GCSF.

Finally, we agree that we should describe detailed experimental proposals for future verification of our hypothesis, as discussed above.

Based on the above discussions, we have revised the whole Discussion section.

More minor points:

page 2, line 22: I think "monotonous" should be "monotonic".

Reply: Thank you. We have revised it.

page 2, line 32: This opening sentence is out of date and will be again when this is published. Everyone knows how important this pandemic is, and this is not needed.

Reply: Thank you. We have deleted it.

page 3, line 52: I appreciate that the authors have worked on interesting methods related to the presentation here, but this model is a standard extension of a virus dynamics model and the reference to physics is not needed.

Reply: We agree that it may appear that our model is an extension of the traditional virus dynamics model, but in fact, not exactly. The most fundamental difference between the cited physical reference and the typical virus dynamics model is that the former postulates the existence of key macroscopic degrees of freedom (called order parameters) of any complex system for each relevant macroscopic function. In many cases, the existence of order parameters is not obvious, but if they do exist (under appropriate statistical average),

then parameters in the model are macroscopic on the scale of the human body and days in time, decoupled from complex microscopic molecular processes. This is the key that our model can systematically depict clinical data. Therefore, the reference of physics is indeed needed.

page 4, Figure 1: Is IL-1 the same as IL-1beta? And why is this mentioned in the legend given that it is not in the model (similarly for macrophages and NK cells)?

Reply: IL-1 beta is one of the IL-1 family. We agree the figure 1 should be consistent with the model; therefore, we have removed the factors in the original figure 1 that is not engaged in the model, see the revised figure 1.

page 5, line 5: I don't see why the inflammation response would follow the immune response. Aren't they part of the same process? This whole paragraph has no citations to back up the choice of mechanisms.

Reply: We agree that the inflammation response is a part of the immune response; thus, we have revised Fig.1 and the corresponding description of IL-6 dynamics in the last paragraph on page 6, as "we propose that IL-6 production by non-neutralizing antibodies dominates the cytokine storm taking place at later stage⁹, and neglect the IL-6 produced by T cells, monocytes and infected cells¹⁰".

Furthermore, we have added the corresponding citations in the " Causal network and mathematic model of the Antiviral-Inflammation responses " section.

page 5, line 42: How about "which incorporates both viral replication and viral clearance..."

Reply: We have revised the text. See the description of the viral dynamics equation, Eq. (3), in the "Model" section.

page 7, line 34: How good a marker is CD3? The description of the T cell types after this was very hard for me to follow.

Reply: CD3+ T cell refers to all T cells (all T cells have CD3 marker). CD8+ T cells are cytotoxic T cells, and when they are activated, they become effector T cells. CD4+ T cells play roles in stimulating and shaping adaptive immune response²³. We have revised the description of effector T cell data in this paragraph as " Because it's difficult to detect the effective T cell number inside organs, we estimate the effector T cell number from the experimentally measured CD3+ T cell data in serum. Zhang et al. ⁵ deduced that the reduction of CD3+ in serum represents the number of T cells that entered organs and produced effector T cells. In this context, we define, at the present stage, the amount of

effector T cell (T_e) is proportional to the reduction of CD3+ in serum (T_{serum}) from its initial value (T_0)."

page 8, line 4: Isn't this just r^2 ? r^2 is not a good statistic for evaluating the quality of a model fit, because it does not take into account the number of parameters.

Reply: The definition of the goodness of fit in the paper is different from R square---we do not use the Total Sum of Squares as the denominator but the maximum of the simulation of the variable. For each patient/group, the goodness of fit of each variable is calculated separately, as shown in Table S9 in the supporting information.

page 8, Figure 2: What is the justification for using T cell data from the severe group for patient 4?

Reply: Zhang et al.⁵ has shown that the CD3+ exhibit clearly a gradual decline as disease deteriorates. Therefore, we assume the T cell data of the patient with the same severity share similarities and approximate the T cell dynamics of the severe patient 4 using the median data of the severe group. We have added the text to the legend of figure 2.

page 8, line 51: Give the equation rather than stating it in words!

Reply: We agree and have revised the text as follows: "we get an analytical solution that predicts the peak is determined from virus inhibition by T cells: $\frac{\tilde{\alpha}^2}{2\beta\delta}$ "

page 10, line 19: This is not a complete sentence, and the "demonstrates the validity" is far too strong a conclusion. The last sentence on this page adds uncertainty to the interpretation of IL-6, which I think needs to be better integrated with the other conclusions.

Reply: We agree that "demonstrates the validity" is too strong and have revised it to be "supports the validity". We have deleted the incomplete sentence and revised the uncertainty text as follows: "The striking feature of non-survivors compared to survivors is the continuous production of IL-6 and organ damage, revealed by zero inhibition rates for all three markers (Figure 5b), while the difference of formation rates of organ damage markers is not remarkable. The zero-inhibition rate of IL-6 for non-survivors may also cause more uncontrolled inflammation before symptom onset so that non-survivors have higher IL-6 than survivors at the 0th day after onset (Figure 4d)."

page 11, line 44: This correlation does not look at all significant, and I do not understand the source of the number of data points of each type.

Reply: It was the bad presentation that masked the significance of the correlation. Therefore, we have revised Figure 5a using statistical values of patients belonging to each severity level, which shows a clearer positive correlation. On the other hand, the Figure 5b shows that when the initial T cell number increases from 0.39 for non-survivors to 0.84 for survivors, the corresponding inhibition rates for IL-6, D-dimer, and highly sensitive cardiac troponin (HSCT) all increase, strongly supporting the positive correlation between the inhibition rate and the initial T cell number.

We have added text in the legend of Figure 5 to explain the source of a number of data points: "a. Positive correlation of T cell's antiviral contribution with initial T cell concentration for mild patients (blue), severe patients (green), survivor group (magenta) and non-survivor group (black). Data points are the means, and error bars are the standard deviations. b. Positive correlation of inhibition rates of IL-6, D-dimer, and high sensitive cardiac troponin (HSCT) with initial T cell concentration for survivor group and non-survivor group. Data points are the means, and error bars are the standard deviations. It is worth mentioning that the three rates of the non-survivor group are all zero and overlap with each other in the figure; The inhibition rates of D-dimer and HSCT of the survivor group are closed and overlap with each other in the figure."

page 11, line 48: Where are regulatory T cells in this model?

Reply: In the updated figure 1, the regulatory T cells play roles in the inhibition rates of IL-6, D-dimer and HSCT, which are assumed as constants during the second stage (the stage that is described by the model, as explained in previous replies), and the impact of regulatory T cells are not explicitly described in the paper.

Figure 5: I mentioned some questions about panel a above, but why are there three symbols in Figure 5b but six items in the legend?

Reply: See the reply above. The three inhibition rates (corresponding to three data points) of the non-survivor group are all zero and overlap with each other in the figure; The inhibition rates of D-dimer and HSCT of the non-survivor group are closed and overlap with each other in the figure. Therefore, there are only three points that can be distinguished in figure 5b.

page 13, line 35: This should be phrased in a more balanced way.

Reply: Yes, we have phrased the sentence as "which complements the complicated multiscale model".

page 15, line 32: I'm really confused by this classification as severe.

Reply: Thank you for pointing this out. We have revised the text to be clearer: "One patient from Kelvin To's cohort has not been identified as severe or critical. Because in the original cohort, it has a low probability to be critical, in this paper, we assign it as severe. "

page 15, line 35: Would the source of data matter for the models? Are T cells and the other markers equally well represented?

Reply: This is an important question. Because it is the macroscopic interaction between virus and immune system that the model describes, the relationship among the evolution of the six elements should be similar in several main organs, such as the nose (nasopharyngeal swab) and oropharynx (oropharyngeal swab). For example, the viral load data of the mild and severe group are from the nasopharyngeal swab, and the data of P1 (mild) and P4 (severe) patients (two individuals) are from the oropharyngeal swab. Figure 2a,e,i, Figure 2 c,g,k, Figure 1S e and k show that the model can describe data from the two sources. The parameters are expected to vary based on different sources. However, It is shown in Figure 3a that the percentage of virus cleared by T cells is 96.72% and 80.59% for the mild and severe group, respectively, and it is 99.43% and 93.84% for P1 (mild) and P4 (severe). Hence, the qualitative conclusion that T cells play a more important role in virus clearance in mild patients than severe patients remains the same in the two parts of the body.

The viral load data and D-dimer data is the most reliable because their samples are from the corresponding organ directly---respiratory tract and blood. The effector T cell data in organs, due to the assumption that the number of effector T cells is proportional to the reduction of CD3+ T cells in the blood, is the least reliable marker. The antibody and IL-6 marker are relatively more reliable than the effector T cell data because they are measured directly in the blood.

page 16, line 10: Why are different scales used?

Reply: The data of non-survivors covers more than one magnitude, so the RSM is used in the log scale. In contrast, the data of survivors are within one magnitude, so the RSM is used in the linear scale. Otherwise, if using the same scale, either the fitting of non-survivors or survivors will look inconsistent with the data.

Reference

1. Perelson, A. S. & Ke, R. Mechanistic Modeling of SARS-CoV-2 and Other Infectious Diseases and the Effects of Therapeutics. *Clin. Pharmacol. Ther.* **109**, 829–840 (2021).
2. Vardhana, S. A. & Wolchok, J. D. The many faces of the anti-COVID immune response. *J. Exp. Med.* **217**, 1–10 (2020).

3. Seow, J. *et al.* Longitudinal observation and decline of neutralizing antibody responses in the three months following SARS-CoV-2 infection in humans. *Nat. Microbiol.* 1–10 (2020). doi:10.1038/s41564-020-00813-8
4. Bjørnstad, O. N., Shea, K., Krzywinski, M. & Altman, N. The SEIRS model for infectious disease dynamics. *Nat. Methods* **17**, 557–558 (2020).
5. Zhang, X. *et al.* Viral and host factors related to the clinical outcome of COVID-19. *Nature* **583**, 437–440 (2020).
6. Mateus, J. *et al.* Selective and cross-reactive SARS-CoV-2 T cell epitopes in unexposed humans. *Science* **370**, 89–94 (2020).
7. Azkur, A. K. *et al.* Immune response to SARS-CoV-2 and mechanisms of immunopathological changes in COVID-19. *Allergy* **75**, 1564–1581 (2020).
8. DeMarco, J. K. *et al.* At the Intersection Between SARS-CoV-2, Macrophages and the Adaptive Immune Response: A Key Role for Antibody-Dependent Pathogenesis But Not Enhancement of Infection in COVID-19. *bioRxiv* 2021.02.22.432407 (2021). doi:10.1101/2021.02.22.432407
9. Zhou, F. *et al.* Clinical course and risk factors for mortality of adult inpatients with COVID-19 in Wuhan, China: a retrospective cohort study. *Lancet* **395**, 1054–1062 (2020).
10. Tay, M. Z., Poh, C. M., Rénia, L., MacAry, P. A. & Ng, L. F. P. The trinity of COVID-19: immunity, inflammation and intervention. *Nat. Rev. Immunol.* **20**, 363–374 (2020).
11. Chakravarty, D. *et al.* Sex differences in SARS-CoV-2 infection rates and the potential link to prostate cancer. *Commun. Biol.* **3**, 1–12 (2020).
12. Parkinson, J. Johnson and Johnson COVID-19 Vaccine is 85% Efficacious in Preventing Severe Disease. *ContagionLive* (2021).
13. Jung, J. H. *et al.* SARS-CoV-2-specific T cell memory is sustained in COVID-19 convalescent patients for 10 months with successful development of stem cell-like memory T cells. *Nat. Commun.* **12**, 1–12 (2021).
14. Leslie, M. T cells found in coronavirus patients ‘bode well’ for long-term immunity. *Science* **368**, 809–810 (2020).
15. Alison Tarke, A. *et al.* Negligible impact of SARS-CoV-2 variants on CD4 + and CD8 + T cell reactivity in COVID-19 exposed donors and vaccinees. doi:10.1101/2021.02.27.433180
16. Tarke, A. *et al.* Impact of SARS-CoV-2 variants on the total CD4+ and CD8+ T cell reactivity in infected or vaccinated individuals. *Cell reports. Med.* **2**, 100355 (2021).
17. Ren, W. *et al.* Research progress of traditional Chinese medicine against COVID-19. *Biomed. Pharmacother.* **137**, 111310 (2021).
18. Xiao, Q. and Jiang, Y. J. and Wu, S. S. and Wang, Y. and An, J. and Xu, W. P. and Wu, J. J. Value analyze of Shufeng Jiedu capsule in treatment of COVID-19. *J. Emerg. Tradit. Chin. Med.* **29**, 756–758 (2020).
19. Ren, J., Zhang, A.-H. & Wang, X.-J. Traditional Chinese medicine for COVID-19 treatment. *Pharmacol. Res.* **155**, 104743 (2020).
20. XIA, L. *et al.* Shufeng Jiedu, a promising herbal therapy for moderate COVID-19: Antiviral and anti-inflammatory properties, pathways of bioactive compounds, and a clinical real-world pragmatic study. *Phytomedicine* **85**, 153390 (2021).
21. Yan, L. & Wu, L. The effect of Taijiquan on the Human Body’s T Lymphocyte Cells and Its Susset. *Wushu Res. public* (2007).
22. Black, D. S. & Slavich, G. M. Mindfulness meditation and the immune system: a systematic review of randomized controlled trials. *Ann. N. Y. Acad. Sci.* **1373**, 13–24 (2016).
23. Bell, L. CD4+ T cells. *British Society for immunology*

Appendix C

Cover response to referees and editors

Thank you for providing feedback for our manuscript. We have replied to the point raised by the associate editor in blue as below:

Associate Editor Comments to Author (Professor Tim Rogers):

Associate Editor

Comments to the Author:

I believe that the authors have addressed the referees comments well everywhere except in the title. Please could they consider changing "is critical" to "may be critical".

Reply: We have revised our title to be “Impairment of T cells’ antiviral and anti-inflammation immunities may be critical to death from COVID-19”.